# MTG8 interacts with LHX6 to specify cortical interneuron subtype identity

Zeinab Asgarian[1,7], Marcio Guiomar Oliveira[1,7], Agata Stryjewska[1,7], Ioannis Maragkos[2], Anna Noren Rubin[1], Lorenza Magno[1], Vassilis Pachnis [3], Mohammadmersad Ghorbani[4,5], Scott Wayne Hiebert[6], Myrto Denaxa[2] & Nicoletta Kessaris [1] ✉

Cortical interneurons originating in the embryonic medial ganglionic eminence (MGE) diverge into a range of different subtypes found in the adult mouse cerebral cortex. The mechanisms underlying this divergence and the timing when subtype identity is set up remain unclear. We identify the highly conserved transcriptional co-factor MTG8 as being pivotal in the development of a large subset of MGE cortical interneurons that co-expresses Somatostatin (SST) and Neuropeptide Y (NPY). MTG8 interacts with the pan-MGE transcription factor LHX6 and together the two factors are sufficient to promote expression of critical cortical interneuron subtype identity genes. The SST-NPY cortical interneuron fate is initiated early, well before interneurons migrate into the cortex, demonstrating an early onset specification program. Our findings suggest that transcriptional co-factors and modifiers of generic lineage specification programs may hold the key to the emergence of cortical interneuron heterogeneity from the embryonic telencephalic germinal zones.

A large diversity of cortical GABAergic interneurons has been identified in the mammalian cerebral cortex on the basis of molecular, electrophysiological, morphological and connectivity signatures[1–5]. The development of this diversity begins early during embryogenesis when cortical interneurons are specified from subcortical precursors[6,7]. Although mature gene networks and epigenetic signatures become evident only when interneurons settle in the cortex[8], specification starts early, and embryonic gene expression in newly differentiated neurons reflects cortical interneuron mature identities[9]. Hence, genetic programs that instruct interneuron fates initiate early but unfold gradually subject to intrinsic and extrinsic influences[10,11].

SST- and Parvalbumin (PV)-expressing cortical interneurons are the two major classes of interneurons that are generated from the MGE[12]. More than 15 different SST and PV interneuron subtypes have been identified to date and yet we lack all knowledge as to how this fine

diversity is generated[4,13]. NKX2-1 and its downstream target, LHX6, constitute the core molecular cascade governing the onset of MGE-interneuron development[14,15]. LHX6 is expressed in the entire MGE cortical interneuron lineage at all prenatal and postnatal stages and is required for specification, migration, laminar distribution and differentiation of MGE interneurons[16–18]. How LHX6 performs these diverse functions remains unknown.

*Myeloid translocation genes* (*Mtg*) encode non-DNA binding transcriptional regulators that associate with DNA-binding factors to recruit regulatory protein complexes to target loci[19,20]. They are so named because they are frequent targets of translocation in acute myeloid leukaemia[20]. MTG proteins are best known as transcriptional co-repressors with high networking capacity that regulate cell fate decisions and differentiation in different systems[21–23]. Three family members have been identified to date: *Mtg8* (*Runx1t1*), *Mtg16* (*Cbfa2t3*)

[1]Wolfson Institute for Biomedical Research, University College London, Gower Street, London WC1E 6BT, UK. [2]Biomedical Sciences Research Center "Alexander Fleming", Vari, Greece. [3]The Francis Crick Institute, London, UK. [4]Centre for Cancer Immunology, Faculty of Medicine, University of Southampton, Southampton General Hospital, Southampton, UK. [5]Department of Human Genetics, Sidra Medicine, Doha, Qatar. [6]Department of Biochemistry, Vanderbilt University School of Medicine, Nashville, TN, USA. [7]These authors contributed equally: Zeinab Asgarian, Marcio Guiomar Oliveira, Agata Stryjewska. ✉e-mail: n.kessaris@ucl.ac.uk

and *Mtgr1* (*Cbfa2t2*)[19,20]. Expression of all three genes has been detected in the central nervous system, but their function remains unknown[24].

We identified *Mtg8* and *Mtg16* as being enriched in MGE-derived cortical interneurons and we explored their function during development. Loss of *Mtg16* was dispensable with regards to specification of MGE interneurons. However, loss of *Mtg8* resulted in autonomous reduction in the number of migrating *Sst* cortical interneurons, a phenotype that resembled *Lhx6* hypomorphic mice. We find that LHX6 and MTG8 proteins interact, and together the two factors are sufficient to promote expression of critical MGE-derived interneuron traits and SST/NPY subtype identity.

## Results

### MTG8 is required for specification of SST interneuron subsets
We detected enriched expression of *Mtg8* (*Runx1t1*) and *Mtg16* (*Cbfa2t3*) in MGE-derived cortical interneurons in a comparative transcriptomic study aimed at identifying MGE and caudal ganglionic eminence (CGE)-derived cortical interneuron gene expression signatures (Supplementary Fig. 1 and Supplementary Data 1). Given the significance of *Mtg* family members in different systems during embryogenesis[21–23] and their detection in different transcriptomic studies[7,25,26], we sought to explore their expression and function in the telencephalon with respect to cortical interneuron development. At E11.5, *Mtg8* transcripts are widely distributed in the subcortical telencephalic mantle whereas *Mtg16* can be detected predominantly in the MGE in a pattern comparable to that of *Lhx6* (Fig. 1a). By E16.5, *Mtg8* is widespread in subpallial and pallial regions, including the cortical plate (CP) and the cortical subventricular and intermediate zones (SVZ/IZ) (Fig. 1b). At this stage, *Mtg16* is present in scattered cells within the developing cortex, in a pattern reminiscent of *Lhx6* (Fig. 1b).

We quantified expression of the two *Mtg* genes in the MGE lineage, using lineage tracing mice[27,28] (Nkx2-1-Cre;R26R-YFP, abbreviated as Nkx2-1Cre;YFP and Lhx6-Cre;R26R-YFP, abbreviated as Lhx6-Cre;YFP) as well as a mouse expressing *β*-galactosidase (*βgal*) under control of *Mtg8* (Supplementary Fig. 2a, b)[29]. A large number of migrating MGE-derived YFP[+ve] cells expressing *Mtg8/βgal/*MTG8 was detected in the developing cortex at E14.5 (Fig. 1c, e and Supplementary Fig. 2e, f). In contrast, only small numbers of MTG16 cells are present in the prospective cortex at this stage (Fig. 1f, h). By E18.5, the relative abundance of *Mtg8/*MTG8 and MTG16 within the cortical MGE lineage was reversed: scarce expression of *Mtg8/*MTG8 was present in MGE interneurons at this stage, whereas MTG16 labelled nearly all MGE-derived cells (Fig. 1d, e, g, h and Supplementary Fig. 2g, h). In addition to interneurons, *Mtg8* was also expressed in pyramidal neurons in the developing CP (Fig. 1b, c, d and Supplementary Fig. 2e, f). MTG16 was restricted to MGE cortical interneurons (Fig. 1g). The dynamic expression of the two *Mtg* genes within the MGE lineage during embryogenesis suggests temporally segregated roles for these two transcriptional regulators in the development of these cells.

In order to assess the requirement for *Mtg8* and *Mtg16* in the development of MGE-derived cortical interneurons, we used mice carrying loss-of-function (LOF) mutations (*Mtg8⁻/⁻* and *Mtg16⁻/⁻*) (Supplementary Fig. 2a–d)[29,30]. At E13.5 *Lhx6*-expressing cortical interneurons migrated normally within the developing cortex in *Mtg8⁻/⁻* and *Mtg16⁻/⁻* mutant embryos (Fig. 2a, c, d and Supplementary Fig. 3a). In contrast, a clear reduction of *Sst* expression was detected in pallial and subpallial regions in *Mtg8⁻/⁻* but not *Mtg16⁻/⁻* embryos (Fig. 2b, c, e and Supplementary Fig. 3b). Within the developing cortex, *Sst* reduction was observed in both the marginal zone (MZ) and the SVZ streams in *Mtg8* mutants (Fig. 2b, c, e). Reduction of *Sst* expression, but not *Lhx6*, was also detected at E18.5 in *Mtg8⁻/⁻* embryos, indicating persistence of the phenotype in late embryogenesis (Fig. 3f, g). The MGE and the developing globus pallidus appeared normal in size in both *Mtg* mutants (Supplementary Fig. 3, Supplementary Fig. 4).

Altogether our data indicate that in the absence of *Mtg8*, *Lhx6*-expressing cortical interneurons migrate into the developing cortex in normal numbers but a significant proportion fails to express *Sst*. The early onset of the phenotype suggests that *Mtg8* is required for *Sst* cell specification early on during development, at the time when these cells emerge from the ganglionic eminence. *Mtg16* is expressed later and is dispensable for *Sst* interneuron generation.

### Autonomous requirement for Mtg8 in interneuron specification
We assessed a range of marker genes expressed in the telencephalon in order to determine whether the loss of *Mtg8* impacted other aspects of the developing telencephalon that may indirectly affect *Sst* specification. At E13.5 we found no change in the expression of neural patterning genes or other key factors expressed in the telencephalon (Supplementary Fig. 4a–o). Notch pathway gene expression was also unaffected in the mutants (Supplementary Fig. 4p–x). Layering of cortical pyramidal neurons was unaltered, although the cortex appeared thinner than that of littermate controls (Supplementary Fig. 4y). In the basal ganglia, there was normal gene expression (Supplementary Fig. 4z). *Mtg8⁻/⁻* embryos lacked a visible anterior commissure (Supplementary Fig. 4z, anterior commissure indicated by an asterisk in control embryos) but showed normal formation of the hippocampal commissure and corpus callosum.

In order to directly determine whether the loss of *Sst* is caused by autonomous defects, we dissociated E13.5 MGE cells from *Mtg8⁻/⁻* or *Mtg16⁻/⁻* embryos and their littermate controls (all labelled with YFP), mixed them together with dissociated WT control MGE cells (labelled by expression of both tdTom and YFP) at defined starting ratios, and transplanted them into host cortices (Fig. 3a). We assessed survival of transplanted cells and expression of SST and PV. *Mtg8* mutant cells showed a ~25% reduction in survival in host cortices 40 days after transplantation (Fig. 3b). A clear decrease in SST cell numbers was observed among transplanted cells from *Mtg8⁻/⁻* embryos (Fig. 3c). This was associated with a small but significant increase in PV cells (Fig. 3d). However, this increase was lost when taking into account the 25% cell death among transplanted cells (see Fig. 3b), indicating that increased abundance of PV was likely due to a smaller surviving transplanted YFP population, presumably caused by cell death of misspecified SST interneurons. These findings confirm that the phenotype observed in germline *Mtg8* mutants is autonomous to the MGE and point to a requirement for MTG8 specifically in SST interneuron development.

### Common putative downstream targets of MTG8 and LHX6
The phenotype observed in *Mtg8* deficient mice, namely, the reduced number of *Sst*-expressing cells but normal number of *Lhx6*-expressing immature neurons migrating into the embryonic cortex, very much resembles the phenotype observed in *Lhx6* hypomorphic mice. In the latter, reduction of LHX6 activity to around 40% caused partial reduction of *Sst* but not PV interneuron numbers in the cortex[31]. The phenotypic similarity between the two mutants prompted us to look for common putative targets that may be regulated by the two factors. We used RNA sequencing (RNA-seq) to profile the transcriptome of purified E14.5 migrating MGE interneurons from embryos lacking either MTG8 or LHX6 and their littermate controls (Fig. 4a). Gene expression analysis identified 268 differentially expressed genes (DEGs) significantly downregulated and 540 genes significantly upregulated in LHX6 mutants compared to controls (Fig. 4a, b) (a full list of genes is shown in Supplementary Data 2). Similar analysis of purified MTG8 mutant cortical interneurons identified 10 genes significantly downregulated and 22 genes significantly upregulated in the mutants (Fig. 4a, b and Supplementary Data 2). Common in both *Lhx6* and *Mtg8* mutant MGE cortical interneurons were 5 downregulated and 3 upregulated genes, with *Sst* and *Npy* being the top hits among the common downregulated genes (Fig. 4c, d). *Thbs1*, *Npy2r* and *Galntl6* were

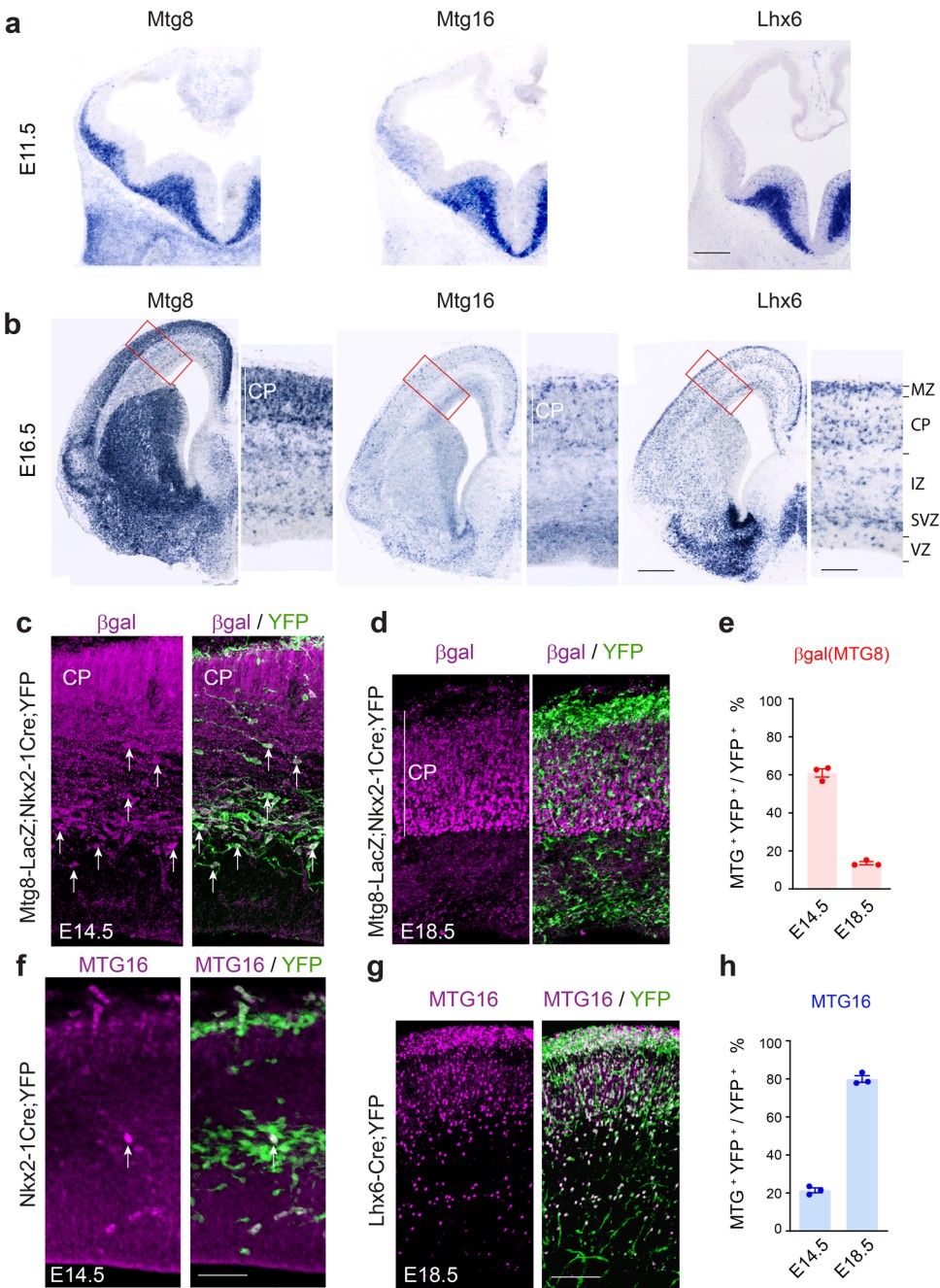

**Fig. 1 | Dynamic expression of *Mtg* genes in the telencephalon.**
**a**, **b** Representative coronal sections through the embryonic telencephalon at E11.5 (**a**) and E16.5 (**b**) showing expression of *Mtg8*, *Mtg16* and *Lhx6*. Expression of *Mtg16* resembles that of *Lhx6*, whereas *Mtg8* is expressed more widely. MZ, marginal zone; CP, cortical plate; IZ, intermediate zone; SVZ, subventricular zone; VZ, ventricular zone. **c**–**e** Expression of *Mtg8* (β-gal) at E14.5 (**c**) and E18.5 (**d**) in the MGE-derived cortical interneuron lineage (Nkx2-1-Cre;YFP) and quantification (**e**). Arrows point to Mtg-expressing cells double labelled for YFP. *n* = 3 embryonic brains at each time-point. **f**–**h** Expression of MTG16 at E14.5 (**f**) and E18.5 (**g**) in the MGE-derived cortical interneuron lineage (Nkx2-1-Cre;YFP) and quantification (**h**). Arrow point to Mtg-expressing cell double labelled for YFP. n = 3 embryonic brains at each time-point. All graphs show mean ± SEM. Source data are provided as a Source Data file. Scale bars: (**a**) 200 μm; (**b**) 400 μm and higher magnification 100 μm; (**c**, **f**) 65 μm; (**d**, **g**) 120 μm.

undetectable by ISH, as was the upregulation of *Sp8*, *Cbln2* and *Pbx3*. The reduction of *Sst* in transcriptomic data is consistent with our previous observations (Fig. 2). We confirmed the downregulation of *Npy* in both *Lhx6* and *Mtg8* mutants (Fig. 4e. Arrowheads point to *Npy* expression in the subcortical telencephalon. *Npy*-expressing migrating interneurons in the cortex are indicated by arrows in e′–e″. Quantification of *Npy* cells migrating within the developing neocortex in areas below the CP is shown). Altogether, our data indicate that *Lhx6* and *Mtg8* are both involved in activating expression of *Sst* and *Npy* (and

possibly other genes), two critical identity factors marking a subpopulation of the SST branch of the MGE lineage.

## MTG8 and LHX6 interact to regulate interneuron identity factors

Given the common putative downstream targets of MTG8 and LHX6, we sought to determine the relationship between the two factors. First, we examined possible genetic interactions. Simultaneous deletion of one allele each of *Mtg8* and *Lhx6* did not cause changes in *Sst*

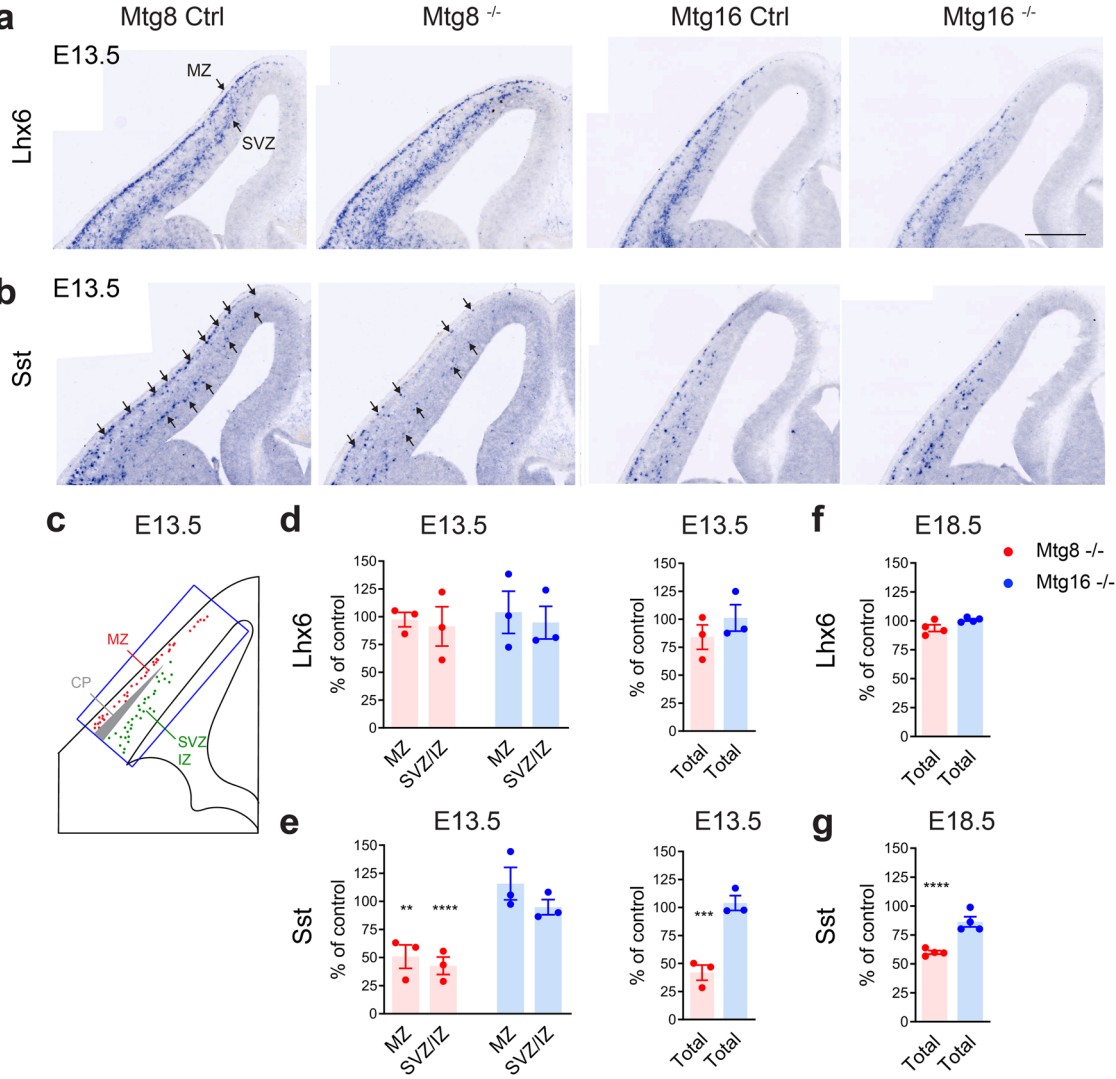

**Fig. 2 | MTG8 is required for embryonic specification of SST interneuron subsets. a**, **b** *Lhx6*- (**a**) and *Sst*-expressing (**b**) MGE-derived cortical interneurons migrating into the developing neocortex at E13.5 in control (Ctrl), *Mtg8*⁻/⁻ and *Mtg16*⁻/⁻ embryos. **c** Scheme showing how *Lhx6* and *Sst* quantification was performed for the data shown in Fig. 2d, e. A rectangle stretching from the cortico-striatal boundary to the tip of the developing neocortex was used to define the area of counting. Migrating interneurons in the MZ and SVZ/IZ as well as total numbers were quantified. The same rectangle was used for all embryos examined in order to allow comparison between control and mutants. **d**–**g** Quantification of cells in the neocortical MZ and the SVZ/VZ, as well as total cell numbers at E13.5 (**d**, **e**) and E18.5 (**f**, **g**). Reduced numbers of migrating *Sst*-expressing cells in *Mtg8* mutant embryos. E13.5: n = 6 Ctrl, n = 3 mutant embryos. E18.5: n = 4 embryos for each genotype. One-way analysis of variance (ANOVA) followed by Uncorrected Fisher's LSD tests. E13.5 *Sst* Mtg8⁻/⁻: MZ P = 0.0083, SVZ/IZ P < 0.0001, Total P = 0.0002. E18.5 *Sst* Mtg8⁻/⁻: Total P < 0.0001. **P < 0.01, ***P < 0.001, ****P < 0.0001. All graphs show mean ± SEM. Source data are provided as a Source Data file. Scale bar: 400 μm.

expression, indicating lack of epistatic interactions between the two genes in the heterozygote state (Supplementary Fig. 5a–f). Complete loss of *Mtg8* did not result in changes in the expression of *Lhx6* (Fig. 2a, d, f) and, conversely, loss of *Lhx6* did not result in loss of *Mtg8* expression (Supplementary Fig. 5g). This indicates that the two genes are activated independently of each other through parallel pathways.

We next used in vitro transfection assays followed by co-immunoprecipitation (co-IP) in order to examine possible protein-protein interactions between LHX6 and MTG8. Co-expression of the two proteins in Cos7 cells followed by immunoprecipitation of LHX6 resulted in robust and consistent co-IP of MTG8, indicating a physical interaction between the two proteins. (Fig. 5a). This interaction was not lost upon deletion of the LIM interaction domains of LHX6 (Fig. 5a). Similar co-IP experiments failed to show an interaction between MTG8 and the LIM-domain binding factor LDB1, a nuclear protein that binds to all LIM-HD factors and is essential for their function (Fig. 5b). These data indicate that MTG8 physically interacts

with LHX6 and this interaction is independent of the LIM domains of LHX6 and the LIM interacting protein LDB1.

In order to test the possibility that MTG8 co-operates with LHX6 to regulate gene targets, we used *ex utero* electroporation to introduce the two cDNAs into the developing neocortex at E14.5-E15.5 (Fig. 5c). On their own, neither MTG8 nor LHX6 were able to promote *Sst* or *Npy* expression in the first 48 h of cortical slice culture following electroporation (Fig. 5d). However, when co-expressed in the developing cortex, the two factors were able to induce robust ectopic expression of *Npy* (Fig. 5d). The absence of *Sst* activation but robust activation of *Npy* upon electroporation of both MTG8 and LHX6 in the cortex persisted even after 7 days of cortical slice culture (Supplementary Fig. 6). The inability of the two regulators to activate *Sst* expression in neocortical progenitors suggests that other factors normally present in ventral telencephalic progenitors may be required to induce its ectopic expression. Altogether, the data point to an obligatory interaction between LHX6 and MTG8 for induction of MGE interneuron critical identity factors.

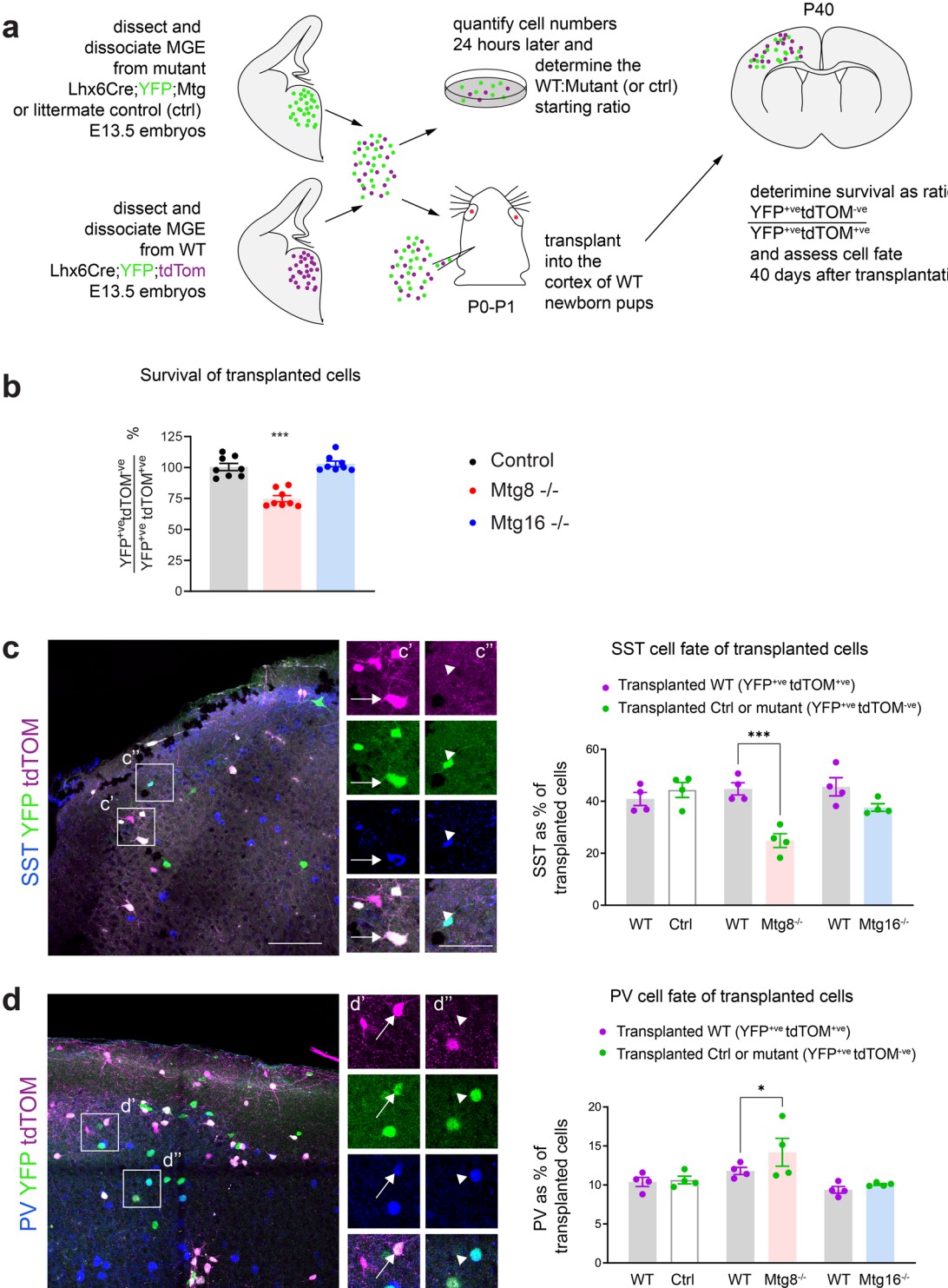

**Fig. 3 | An autonomous requirement for MTG8 for embryonic specification of SST interneuron subsets. a** Experimental design for in vivo transplantation of MGE precursors from control and mutant embryos at E13.5 into the cortex of newborn pups. **b** Quantification of surviving transplanted cells within the somatosensory cortex 40 days post transplantation. *n* = 8 brains for each genotype. Two-tailed Mann–Whitney *U* test. ****P* < 0.001.

**c, d** Immunohistochemistry for SST (**c**) or PV (**d**) and tdTOM/YFP to detect transplanted control and mutant MGE precursors differentiating into SST or PV interneurons and quantification. *n* = 4 brains per genotype. Two-way ANOVA followed by Uncorrected Fisher's LSD tests. **P* = 0.0307, ****P* < 0.001. All graphs show mean ± SEM. Source data are provided as a Source Data file. Scale bars: (**c**, **d**) 100 μm and higher magnification 50 μm.

We next tested the capacity of MTG8 to promote SST-NPY interneuron fates within the MGE lineage by overexpressing the gene in the entire lineage. We dissociated E13.5 MGE cells from Nkx2-1-Cre;tdTomato mice and transduced them with a lentivirus

expressing a Cre-conditional MTG8-2A-VENUS cDNA before transplanting them into P0 WT host cortices (Fig. 6a and Supplementary Fig. 7a, b). Host brains were examined 40 days after transplantation for expression of SST, NPY and PV. Non-transduced tdTomato

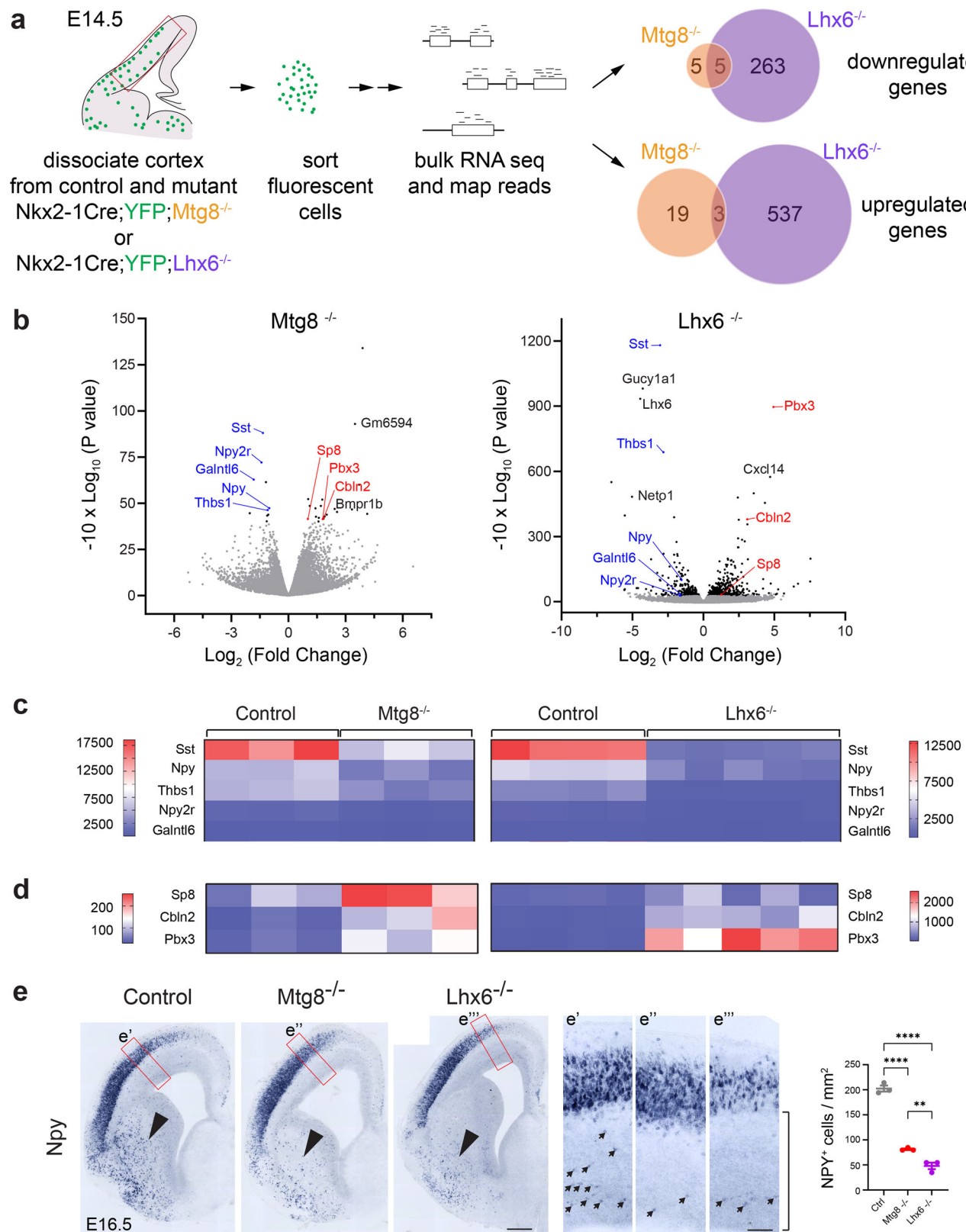

+veVenus-ve transplanted MGE cells were used as controls. Over-expression of MTG8 in MGE lineage cells resulted in a small but significant increase in the percentage of SST+ve cells (~14% increase). NPY+ve cells more than doubled in numbers among the transduced population (~140% increase). These changes were accompanied by a small but significant decrease in PV+ve cells (~35% decrease) (Fig. 6b).

These findings suggest that MTG8 expression is sufficient to promote SST and NPY interneuron fates at the expense of PV fates in a subset of MGE progenitors. Whether the reduction in PV cells reflects reduced viability of prospective PV cells when MTG8 is overexpressed or a re-specification of PV cells into SST interneurons remains unknown.

**Fig. 4 | *Sst* and *Npy* are common target genes downstream of MTG8 and LHX6. a** Experimental design for identification of downstream genes requiring MTG8 and/or LHX6 function within the MGE-derived cortical interneuron lineage using transcriptomic profiling of purified cells from E14.5 neocortex. **b** Volcano plots showing statistical significance ($-10 \times \log_{10} P$ values) against fold change ($\log_2$ fold change) between control and mutant MGE-derived cells for *Mtg8* and *Lhx6*. Negative $\log_2$ fold change values indicate decreased expression in mutants compared to controls. Positive $\log_2$ fold change values indicate increased expression in mutants compared to controls. DEGs with a padj <0.05 are shown in black/blue/red. Common DEGs in both *Mtg8* and *Lhx6* mutants are indicated in blue (downregulated) and red (upregulated). **c, d** Heatmaps showing common downregulated (**c**) and common upregulated (**d**) genes in *Mtg8* and *Lhx6* mutant animals. *Sst* and *Npy* are among the top downregulated genes in both mutants. **e** Representative coronal sections through the embryonic telencephalon at the level of the developing septum at E16.5 showing expression of *Npy* in control, *Mtg8*$^{-/-}$ and *Lhx6*$^{-/-}$ mutant animals. An area of the developing neocortex is shown at higher magnification in e'–e'''. Arrows indicate migrating *Npy*-expressing cortical interneurons in the SVZ/IZ. Quantification represents Npy$^{+ve}$ cells in the developing neocortex in the SVZ/IZ areas underneath the CP (area indicated in e''). Reduced expression of *Npy* can be seen in the developing basal ganglia (arrowheads) and neocortex (arrows) of both mutants. $n = 3$ embryonic brains for each genotype. One-way ANOVA followed by Uncorrected Fisher's LSD tests. ****$P < 0.0001$, **$P = 0.0042$. Graph shows mean ± SEM. Source data are provided as a Source Data file. Scale bars: 250 μm and higher magnification 60 μm.

## Essential role for Mtg8 in SST-NPY interneuron development

Constitutive *Mtg8* mutant mice die soon after birth due to gastrointestinal defects[29]. Therefore, we generated mice carrying a Cre conditional allele for *Mtg8*, in order to assess the postnatal development of the MGE lineage and the loss of SST interneuron subsets (Fig. 7a and Supplementary Fig. 7c, d). We first examined these mice at embryonic stages. At E13.5, expression of *Lhx6* in pallial and subpallial regions of the telencephalon in *Mtg8* cKO embryos (cKO, Nkx2-1-Cre;Mtg8$^{fl/fl}$) appeared normal compared to controls (Ctrl, Mtg8$^{fl/fl}$) (Supplementary Fig. 8a). In contrast, there was a visible reduction in *Sst* cells in the subcortical and cortical mantle in cKO embryos (Supplementary Fig. 8b). This was consistent with our observations in germline *Mtg8* knock-out mice (Fig. 2a–e and Supplementary Fig. 3a, b). Expression of SOX6, another key determinant of telencephalic *Sst* interneurons was unaffected in the absence of MTG8 indicating that, like LHX6, SOX6 is activated independently of MTG8 (Supplementary Fig. 8c).

We quantified cortical interneurons in adult *Mtg8* cKO mice in order to determine the postnatal effects of MTG8 loss on the MGE lineage. A marked decrease in cortical *Sst* interneuron numbers was observed in the lower layers of the cortex in *Mtg8* cKO mice (Nkx2-1-Cre;Mtg8$^{fl/fl}$) (Fig. 7d, e). Unlike our observations at embryonic stages, this was accompanied by a loss of *Lhx6*-expressing cells (Fig. 7b, c). This may reflect reduced survival of mis-specified SST cortical interneurons lacking MTG8, in line with the transplantation experiments where loss of MTG8 caused reduced survival of transplanted MGE cells (Fig. 3b). *Pv* interneuron numbers are normal in the absence of *Mtg8* (Fig. 7f, g). Late deletion of MTG8 using Lhx6-Cre, which recombines during the migration of cortical interneurons[27], resulted in normal numbers of cortical *Sst, Pv* and *Lhx6* interneurons (Supplementary Fig. 9a), confirming an early requirement for MTG8 in the development of the lineage. A reduction of *Sst* but not *Pv* numbers upon early deletion of the gene in the MGE lineage was also observed in the hippocampus CA1 area and striatum of *Mtg8* cKO mice compared to controls (Supplementary Fig. 9b, c), indicating a widespread loss of *Sst* subsets throughout the anterior forebrain.

In order to determine whether there was a subtype-specific loss of SST interneurons in the absence of MTG8, we quantified SST$^{+ve}$ cortical interneurons co-expressing nNOS or CR, as well as SST cells co-expressing or lacking NPY (Fig. 8a). MGE-derived CR-expressing cells (all of which express SST), as well as SST interneurons co-expressing nNOS, were unaffected in the absence of MTG8 (Fig. 8b, c). In contrast, a significant reduction of SST interneurons co-expressing NPY was observed in the absence of MTG8 (Fig. 8d). Cortical SST interneurons lacking NPY were unaffected (Fig. 8e). Striatal interneurons co-expressing SST and NPY were also reduced in cKO animals compared to controls, whereas SST$^{+ve}$nNOS$^{+ve}$ striatal interneurons were unaffected (Supplementary Fig. 9c). Altogether our data demonstrate that MTG8 interacts with LHX6 in embryonic MGE progenitors to promote SST$^{+ve}$NPY$^{+ve}$ telencephalic interneuron subtype identity.

## Discussion

We identified *Mtg8* and *Mtg16* as being expressed in the MGE lineage of cortical interneurons and we demonstrated a requirement for MTG8 in the development of SST-NPY interneuron subtypes. MTG8 is activated independently of LHX6 and is widely expressed in the ganglionic eminences. The two transcriptional regulators, LHX6 and MTG8, are co-expressed in the MGE where they interact to promote expression of critical cortical interneuron subtype identity genes. The SST-NPY cortical interneuron fate is initiated early, well before interneurons migrate into the cortex, demonstrating an early onset specification program.

LHX6 is the earliest known transcriptional regulator that marks all MGE-derived interneurons as soon as they emerge from the VZ. It is pivotal for the development of the lineage, with multiple functions ranging from migration to identity specification[16–18]. Our knowledge into the mechanism of action of LHX6 is very limited and, to date, binding partners of LHX6 have been entirely elusive. Intriguingly, reduced levels of LHX6, caused by expression of a hypomorphic *Lhx6* allele, cause loss of only subsets of SST interneurons, suggesting a dose-dependent requirement for specification of different branches of the MGE lineage[31].

Our data demonstrate a clear role for MTG8 in the specification of SST interneuron subsets. *Mtg8* is expressed in the SVZ and postmitotic mantle of the MGE, where expression overlaps with *Lhx6*. *Mtg8* and *Lhx6* are activated independently of each other and may act through parallel pathways to specify SST interneuron fates. Alternatively, MTG8 and LHX6 proteins may interact to mediate SST interneuron specification. Both scenarios are compatible with (a) the phenotypic similarity between the *Lhx6* hypomorphic mice and the *Mtg8* LOF mice, i.e. partial loss of *Sst* and *Npy* expression in migrating interneurons upon reduction/loss of either factor, (b) the identification of *Sst* and *Npy* as putative common downstream targets of LHX6 and MTG8 and (c) the ability of the two proteins to induce ectopic expression of *Npy* upon co-expression in cortical progenitors. However, the finding that the two proteins physical interact when co-expressed in vitro indicates that MTG8 forms an interacting partner of LHX6. The two factors together promote the emergence of the SST-NPY interneuron identity in SVZ/early postmitotic cells prior to their migration into the developing cortex. This interaction between LHX6 and MTG8 provides insight into the mechanism of action of LHX6 and lineage divergence within the developing MGE.

The function of MTG8 within the LHX6-MTG8 complex remains to be explored. One possibility is that MTG8 stabilizes LHX6 onto target DNA, thereby increasing its effective activity. This would be consistent with the phenotype similarity between MTG8 null mice and LHX6 hypomorphs. Such a function has recently been proposed for another MTG family member, MTGR1, during germline development[21]. Just like MTG8, MTGR1 lacks intrinsic DNA binding capacity but is thought to stabilize key DNA-binding factors onto chromatin via its oligomerization, increasing their 'on rate' and functional efficiency[21]. Alternatively, or, in addition to increasing the stability of LHX6 onto chromatin, MTG8 might interact with transcriptional co-activators or co-repressors, such as the Sin3A/HDAC[32] and LIM-HD family members[33] to regulate

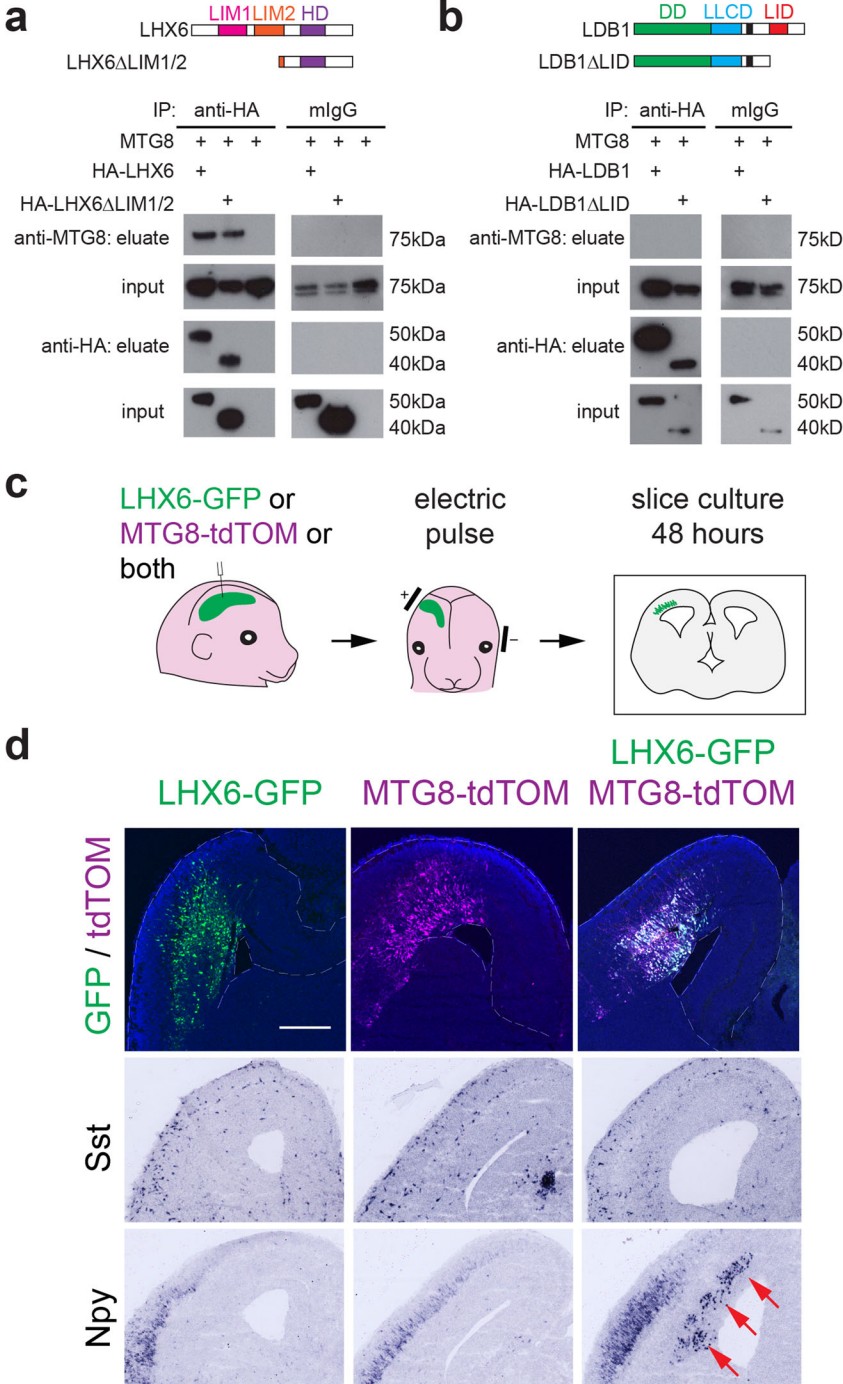

**Fig. 5 | MTG8 interacts with LHX6 to promote SST-NPY fate. a** Domain structure of LHX6 and the truncated LHX6 protein used in this study, LHX6ΔLIM1/2, which lacks both LIM domains. Immunoprecipitation of WT or truncated LHX6 results in co-IP of MTG8 in transient transfection assays in Cos cells. Uncropped blots are provided as a Source Data file. **b** Domain structure of LDB1 and the truncated LDB1 protein used in this study, LDB1ΔLID, which lacks the LIM-interaction domain. Immunoprecipitation of WT or truncated LDB1 does not result in co-IP of MTG8 in transient transfection assays in Cos cells. Uncropped blots are provided as a Source Data file. **c** Experimental scheme for ectopic expression of LHX6-GFP, or MTG8-tdTOM, or both, in the embryonic cortex via whole head electroporation at E14.5-E15.5, followed by slice culture of coronal sections from electroporated brains. **d** Expression of *Sst*, and *Npy* in the developing neocortex upon ectopic expression of MTG8 and/or LHX6. *Npy* is upregulated only in the presence of both, MTG8 and LHX6 (arrows). Scale bar 400 μm.

downstream targets. Both, activating, as well as repressing functions have been proposed for LHX6[15]. MTG proteins are transcriptional regulators with high networking capacity enabled by their multiple protein interaction domains[19]. Partners of MTG proteins include basic helix loop helix factors in the nervous system and in erythroid progenitors[34–37]. During erythroid differentiation, MTG16 associates with the LDB1 complex to maintain progenitors in a primed state[22,38]. Further work is

needed to identify other components of the LHX6-MTG8 complex that mediate cortical SST-NPY interneuron specification.

Embryos lacking MTG8 show normal expression of *Lhx6* in the developing cortex but, by P40, a marked reduction of *Lhx6*-expressing cells is observed. This may reflect eventual cell death of mis-specified MGE interneurons lacking MTG8 and is consistent with our observation of cell loss upon transplantation of MGE progenitors lacking

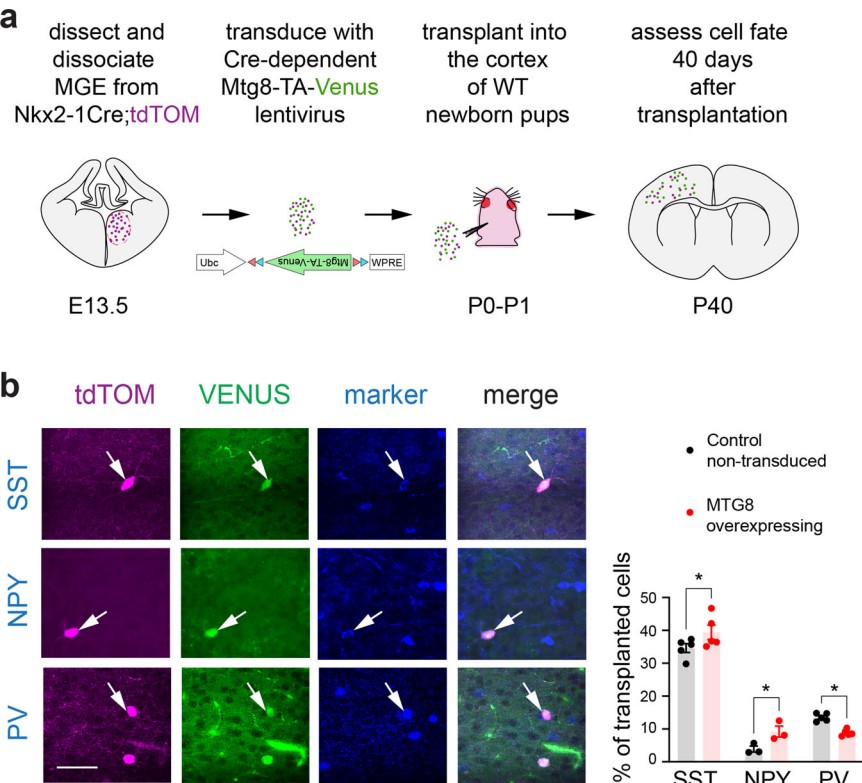

**Fig. 6 | MTG8 promotes SST-NPY fate in the MGE lineage. a** Experimental scheme for overexpression of MTG8 in the MGE lineage by lentiviral transduction of dissociated MGE precursors, followed by transplantation into the cortex of newborn pups. Analysis was carried out in the somatosensory cortex 40 days post transplantation. **b** Immunohistochemistry and quantification for SST, NPY or PV together with tdTOM and VENUS to detect transplanted and/or transduced MGE precursors differentiating into different interneuron subtypes. A minimum of 300 transplanted cells per experiment were counted. $n = 5$ independent transplantations for SST and PV, $n = 3$ independent transplantations for NPY. Two-Way ANOVA was followed by Uncorrected Fisher's LSD tests. SST $P = 0.0147$, NPY $P = 0.0319$, PV $P = 0.0166$. *$P < 0.05$. Graph shows mean ± SEM. Source data are provided as a Source Data file. Scale bar 25 μm.

functional MTG8. The eventual loss of *Mtg8* mutant cortical interneurons is pointing to an essential requirement for MTG8 that may or may not extend beyond expression of the two critical identity genes SST and NPY. Additional characterisation of cortical interneurons lacking MTG8 at postnatal stages is needed to determine whether other aspects of SST interneurons are affected by the loss MTG8.

While *Mtg8* clearly has an early role in MGE cortical interneuron development, *Mtg16* is activated later and is dispensable for early specification events. MTG16 is expressed in the entire MGE lineage. Mouse mutants in MTG16 have normal numbers of cortical interneurons but develop occasional seizures at 6 months (our observations). Double MTG8/MTG16 mutant MGE progenitors transplanted into the newborn cortex do not show additive phenotypes in terms of cortical interneuron numbers, indicating lack of compensatory functions between the two genes (our observations). The functions of MTG16 remain unknown but it may represent an example of a cortical interneuron development program that initiates early in the MGE and unfolds gradually, after interneurons reach the cortex.

Human mutations in *MTG8* have been reported and include a deletion within the *MTG8* gene (known as *RUNX1T1*) and a translocation that disrupts the *MTG8* locus[39,40]. In both cases the mutations are associated with intellectual disability/mental retardation. Learning disabilities in these patients occur early during childhood thus pointing to neurodevelopmental defects arising through loss of MTG8 function.

The timing of neuronal diversification has been studied for many years. Recent data from the developing cortex have found that neuronal diversification occurs at the level of post-mitotic neurons and not progenitors[41]. Similarly, there is no evidence for neuronal divergence prior to cell cycle exit in the ganglionic eminences, and prediction of SST versus PV cell identity, based on whole transcriptome and epigenome content, is only possible at early postnatal stages[8]. However, PV and SST cortical interneurons clearly diverge well before that, and immature SST interneurons can be seen migrating into the cortex at embryonic stages. This early divergence between PV and SST interneurons is discernible through single cell transcriptomic studies of embryonic neurons from mice[6,7,42] and humans[43,44]. Our data support a model whereby even fine subclasses of interneurons are specified early, soon after exit from the VZ, and before young neurons reach the cortex. The fine-scale specification may occur, not through novel, as yet unidentified genetic pathways, but rather through the modification of generic MGE programs by the formation of subtype-specific protein complexes. Identifying such complexes may hold the key to understanding cortical interneuron lineage specification and divergence.

SST interneurons represent ~40% of all MGE-derived interneurons[27] and include at least 40 different transcriptomic subtypes (https://portal.brain-map.org/). It is unknown what the functions of all the different subtypes are. SST-NPY neurons represent around half of the SST population and, apart from a small subset of long-projecting neurons that co-expresses *Nos1/Chodl*[4,13] and are active during sleep[45], we have no further knowledge into the functions of this large neuronal population. As our ability to identify specific subtypes of neurons in live behaving animals improves, so will our understanding of the function of cortical interneuron diversity and the contribution of subtype-specific defects to human interneuron-based disorders.

## Methods
### Transgenic mice
*Mtg8*-LacZ KI mouse (MGI:2183012)[29], *Mtg16* mutant (MGI:3815201)[30], *Lhx6* floxed mice (MGI:6466660), *Nkx2.1-Cre* (MGI:3761164)[28], *Lhx6-Cre*

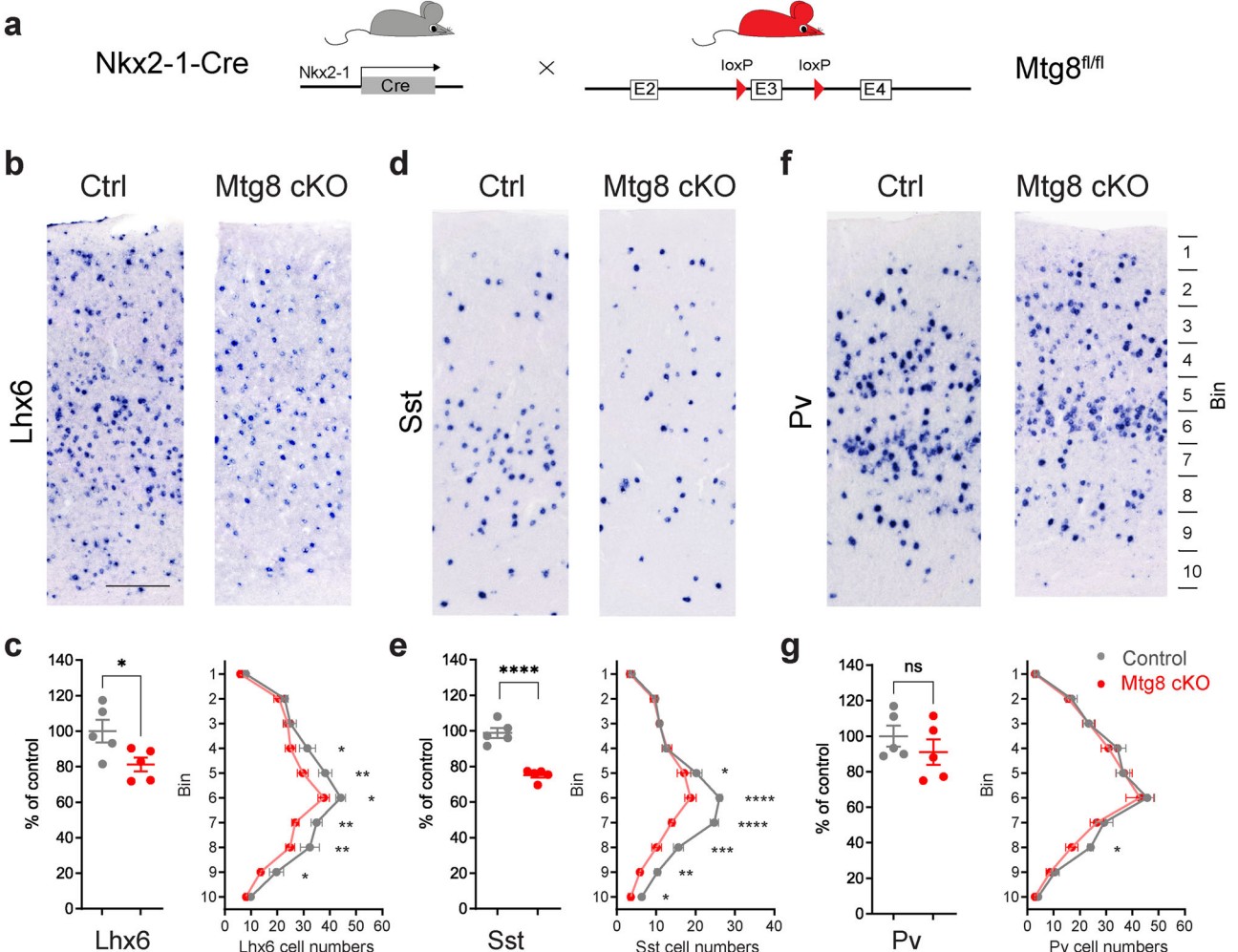

**Fig. 7 | Mtg8 is required for specification of cortical SST interneuron subsets.**
**a** Strategy for the generation of *Mtg8* conditional mutant animals in the MGE using Nkx2-1Cre. **b–g** RNA in situ hybridization detecting *Lhx6, Sst* and *Pv* in the cortex of Ctrl and *Mtg8* cKO animals at P40 (**b, d, f**) and quantification of total numbers and in different bins along the dorso-ventral extent of the somatosensory cortex (**c, e, g**). *n* = 5 brains per genotype. Total numbers: two-tailed unpaired *t* tests. Lhx6

*P* = 0.0369, Sst *P* < 0.0001. Bin distribution: two-way RM ANOVA followed by Uncorrected Fisher's LSD tests. Lhx6: level 4 *P* = 0.0233, level 5 *P* = 0.003, level 6 *P* = 0.022, level 7 *P* = 0.0049, level 8 *P* = 0.0083, level 9 *P* = 0.0345. Sst: level 5 *P* = 0.0245, levels 6 and 7 *P* < 0.0001, level 8 *P* = 0.0001, level 9 *P* = 0.0017, level 10 *P* = 0.0449. Pv: level 8 *P* = 0.0386. *\*P* < 0.05, *\*\*P* < 0.01, *\*\*\*\*P* < 0.0001. All graphs show mean ± SEM. Source data are provided as a Source Data file. Scale bar 200 µm.

(JAX 026555)[46,27], *Nestin-Cre* (JAX 003771), *Dlx1-lox-Venus-lox*[19,20] and two reporter mice, Rosa26R-YFP (JAX 006148)[47] and Rosa26R-tdTomato (JAX 007914)[48] were used in this study. The *Lhx6* conditional mutant mice were used to generate the germline loss-of-function allele used in this study. The *Mtg8* conditional mutant mice (C57BL/6N-Runx1t1^tm1a(EUCOMM)Hmgu/H) were generated by the IMPC and the Mary Lyons Centre at UKRI-MRC Harwell. All mice used in this study were maintained on a C57BL/6/CBA mixed background. Male and female mice were used in this study. Mouse colonies were maintained at the Wolfson Institute for Biomedical Research, University College London, in accordance with United Kingdom legislation (ASPA 1986), and at the BSRC "Al. Fleming", according to the European Union ethical standards outlined in the Council Directive 2010/63EU of the European Parliament on the protection of animals used for scientific purposes.

**Plasmid construction**
A full list of plasmids used in this work is found in Supplementary Data 3. Plasmids generated for the purposes of this study are described below.
*pCMV3-Myc-HA-HA-LHX6* used in co-IP experiments: the coding sequence of mouse *Lhx6* (ENSMUST00000112961) lacking an initiation

codon, was cloned in-frame into the pCMV3-Myc-HA-HA vector. The pCMV3-Myc-HA-HA vector was generated by substituting the pCMV promoter region from the pCMV3 vector with a 2.1 kb fragment containing the pCMV3-1/E1 promoter. One Myc tag and two HA tags were subsequently added in-frame.
*pCMV3-Myc-HA-HA-LHX6ΔLIM* used in co-IP experiments: it is based on pCMV3-Myc-HA-HA-LHX6 but lacks sequences coding for functional LIM domains.
pCAGGS-LHX6-IRES-GFP[49] was used in ex-utero gain-of-function experiments.
*pCAGGS-MTG8*: used in co-IP experiments: The *Mtg8* coding sequence from the cDNA clone FANTOM 6332428O11 was cloned downstream a CAGGS promoter and upstream an SV40 polyadenylation signal sequence in pBluescript.
*pCAGGS-MTG8-IRES-tdTomato* used in ex-utero gain-of-function experiments: The *Mtg8* coding sequence from the cDNA clone FANTOM 6332428O11 was cloned downstream a CAGGS promoter and upstream an IRES-tdTomato-SV40 polyadenylation sequence in pBluescript.
*pUbc-Dio-MTG8-2TA-VENUS-WPRE* used to generate Lentiviral particles for transfection experiments: The *Mtg8* coding sequence without the stop codon was amplified from a cDNA clone (FANTOM

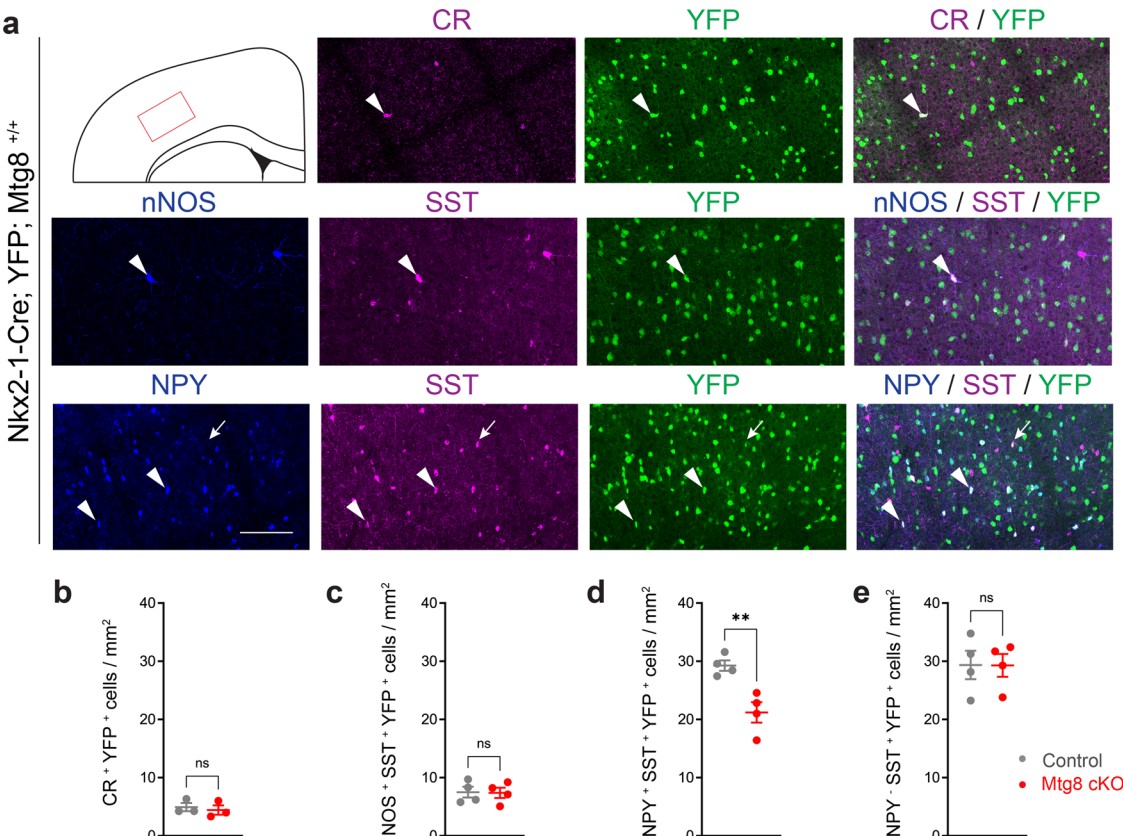

**Fig. 8 | Mtg8 is required for specification of cortical SST-NPY interneuron subsets. a** Immunohistochemistry detecting co-expression of CR/YFP or nNOS/SST/YFP or NPY/SST/YFP in the somatosensory cortex in control animals at P40. Arrowheads point to double-labelled cells for CR/YFP and triple-labelled cells for nNOS/SST/YFP and NPY/SST/YFP. Arrows point to double-labelled SST/YFP cells. **b**–**e** Quantification of the above co-expression profiles in control and *Mtg8* cKO animals. In the absence of MTG8 there is a significant reduction of SST interneurons co-expressing NPY. All other SST subtypes examined remain unaffected. CR+YFP+ *n* = 3 brains per genotype. nNOS+SST+YFP+, NPY+SST+YFP+ and NPY-SST+YFP+ *n* = 4 brains per genotype. Two-tailed unpaired *t* tests. **\*\*P* = 0.0064. All graphs show mean ± SEM. Source data are provided as a Source Data file. Scale bar 130 μm.

6332428O11) and cloned in-frame with a T2A sequence followed by Venus. This sequence (referred to as Mtg8-TA-Venus) was flanked by antiparallel lox sites (lox2272 and loxP) from RV-Cag-Dio-GFP (addgene #87662) and cloned in an inverted orientation into the lentiviral vector FUtdTW (addgene #22478) replacing the tdTomato sequence.

## Tissue preparation
The day of the vaginal plug was considered E0.5, and the day of birth was considered day 0. Whole embryo heads (for embryos E13.5 and younger) or embryonic brains (for embryos older than E13.5), were fixed overnight in 4% (w/v) paraformaldehyde (PFA) in PBS. Postnatal animals were anesthetized and perfused first with saline (0.9% NaCl) followed by 4% (w/v) PFA through the left ventricle of the heart. Adult brains were dissected out, sliced into 2 or 3 mm-slices using a mouse brain coronal matrix (PlasticsOne), and post-fixed in 4% PFA overnight. Fixed samples were cryoprotected overnight by immersion in 20% (w/v) sucrose in PBS. Samples were embedded in Tissue-Tek OCT compound (R. A. Lamb Medical Supplies, Eastbourne, UK), frozen on dry ice, and stored at −80 °C. Post-transplantation animals were treated with 4% PFA overnight, as described above, brains were removed and either sectioned on a vibratome at 50 μm and the sections stored at 4 °C in PBS long-term or stored intact in PBS long-term. Tissue preparation for immunohistochemistry with the goat anti-MTG8 or the rabbit anti-MTG16 antibody involved fixation for 1 hour in 1X MEMFA fixative (0.1 M MOPS, 2 mM EGTA, 1 mM MgSO4, 3.8% Formaldehyde), followed by embedding in Tissue-Tek OCT compound.

## Immunohistochemistry
Embryonic brains were cut on a cryostat into 20-μm-thick coronal sections and collected directly on Superfrost plus slides (BDH Laboratory Supplies, Poole, UK). Adult sections were cut coronally (30 μm thickness) and were serially collected in PBS for "free floating" procedure. All sections were blocked in PBS containing 10% heat-inactivated serum (sheep or donkey) (Sigma, St. Louis, MO) and 0.1% Triton X-100 at room temperature for 1 hour. Immunohistochemistry was performed with the following primary antibodies: rat anti-GFP IgG2a (1:1000; Nacalai Tesque, Kyoto,Japan, #04404-84); rabbit anti-GFP (1:5000; Abcam, UK, #ab290); chicken anti-GFP (1:500; Aves Labs, #GFP-1020); rabbit anti-β-galactosidase (1:2000, MP Biomedicals, # 0856032); goat anti-MTG8 (1:200, Santa Cruz Biotechnology, #sc-9737); rabbit anti-MTG16 (1:300, Abcam, UK, #ab33072); rabbit anti-PV (1:1000, Chemicon Millipore, #MAB1572); mouse anti-PV (1:1000, Swant Bellizona Switzerland, #235); rabbit anti-SST (1:200, Peninsula Labs, #T-4103.0050); rat anti-SST (1:500, Chemicon Millipore, #MAB354); rabbit anti-calretinin (1:1000, Swant Bellizona Switzerland, #7697); mouse anti-CR (1:500, Swant Bellizona Switzerland, #6B3); rabbit anti-NPY (1:1000, ImmunoStar, #22940), sheep anti-NPY (1:500; Abcam, UK, # ab6173); rabbit anti-nNOS (1:1000; Immunostar, #24287); rabbit anti-SOX6 (1:500, Abcam #ab30455). Primary antibodies were applied overnight at 4 °C. Secondary antibodies used were raised in donkey and were conjugated with AlexaFluor 488, AlexaFluor 568, and AlexaFluor 647, all used at 1:1000; (Invitrogen, Carlsbad, CA) for 60 min at room temperature together with Hoechst 33258 (1:1000; Sigma) to detect cell nuclei. All secondary antibodies were diluted in

block solution (1:1000). Floating sections were transferred onto Superfrost plus slides (BDH Laboratory Supplies) and air dried. All sections were coverslipped with Dako fluorescent mounting medium.

### RNA In situ hybridization (ISH) and fluorescent ISH (FISH)

Tissue preparation and RNA ISH was carried out as previously described[50]. Gene expression was examined using DIG-labelled RNA probes followed by detection with sheep anti-Digoxigenin Fab fragments conjugated with alkaline phosphatase (Sigma-Aldrich, #11093274910) and chromogenic development of the signal using NBP (Generon, #10008) BCIP (Generon, #21530050) according to manufacturer's instructions. Plasmids used to generate RNA probes for ISH are indicated in Supplementary Data 3.

For FISH, DIG-labelled probes were detected using sheep anti-Digoxigenin Fab fragments conjugated with peroxidase (Sigma-Aldrich, #11207733910) and developed using the TSA Plus Cyanine 3 signal amplification system (PerkinElmer, #NEL744001KT). FISH was sometimes followed by immunofluorescent detection of protein expression using standard protocols (see Immunohistochemistry).

### Image processing

Images were captured with a Zeiss fluorescent microscope or with a Leica confocal microscope. Images were further processed with Adobe Photoshop (Adobe Systems Inc., San Jose, CA) for general contrast and brightness enhancements. The final compositing of the figures was performed with Adobe Illustrator (Adobe Systems Inc., San Jose, CA). Images of RNA ISH were taken using a ZEISS Axio Scan.Z1 and processed using ZEISS ZEN lite software.

### Lentiviral transfection and in vivo transplantation

Dissociated MGE cells were incubated with concentrated virus (prepared using standard protocols) at 37 °C for 45 min in NB/B27 medium containing Polybrene. Cells were resuspended in NB/B27 medium and transplanted into the cortex of P0-P1 pups as previously described before[51]. Briefly, glass needles for injection were pulled using a Sutter micropipette puller and cut to approximately 100 μm diameter under a dissecting microscope and sharp scissors. The needles were loaded onto a CellTram Vario manual microinjector and back-filled with mineral oil. Cells were gently aspirated into the needle and the needle was placed on a stereotaxic frame. P0-P1 pups were anaesthetized on ice for 2–4 min and immobilised onto the stereotaxic frame using adhesive tape. Each pup received 4–8 injections of 70–100 nl cell suspension on both hemispheres. Pups were then placed on a warm pad and following recovery returned to the nest.

### Dissection, Fluorescence-activated cell sorting (FACS) and RNA extraction from cortical cells for microarray analysis

Cortical tissue was dissected from embryos of the following genotypes: Nkx2-1-Cre[Tg];R26R-YFP[fl/fl] and Nkx2-1-Cre;Dlx1-lox-Venus-lox[19,20]. Samples were pooled into Earle's Balanced Salt Solution (EBSS, Invitrogen) and digested with 2.5 units/ml Papain (Worthington) supplemented with 0.1 mg/ml DNase I (Sigma Aldrich, #D5025-15KU) in a final volume of 1 ml per 2–3 brains. The tissue was dissociated by gentle trituration, before centrifugation and resuspension cold Optimem™ I reduced serum medium (Invitrogen) for FACS. Cells were excited with a 488 nm laser, and fluorescence was detected with 530/40 nm and 580/30 nm bandpass filters. Single, whole cells were isolated from debris based on side scatter and forward scatter parameters. Background fluorescence was determined based on the non-fluorescent control samples, and cells from the upper 80% of non-background events were collected as positive fluorescent cells. Fluorescent cells were sorted directly into tubes prefilled with 500 μl Buffer RLT (lysis buffer, RNeasy Micro Kit, Qiagen). Where the collected volume exceeded 1 ml, a second collection tube was used, ensuring a maximum 1:3 dilution of Buffer RLT. The cells were homogenized by passing five times through a sterile 21 G needle fixed to a 2 ml syringe, and frozen at −80 °C for later RNA extraction using RNeasy Plus Micro Kit (Qiagen, #74034) according to the manufacturer's instructions. RNA quality was assessed using TapeStation RNA ScreenTape (Agilent) according to manufacturer's manual.

### Microarray analysis

cDNA was amplified from extracted RNA using the NuGEN® Ovation Pico System. For each microarray, 4 μg of cDNA was labelled and hybridized to an Affymetrix Mouse Whole Genome 430 2.0 array, with three independent replicates per condition. The Bioconductor software package was used to background-correct and normalize the data, and to calculate mean fold change (FC) and false discovery rates (FDR) for various pair-wise comparisons (e.g., E14.5 MGE versus E16.5 CGE). Heatmaps were created using GraphPad Prism software.

### Dissection, Fluorescence-activated cell sorting (FACS) and RNA extraction from cortical cells for RNAseq

Cortical tissue was dissected from E14.5 embryos of the following genotypes: Mtg8 control (Mtg8[+/+];Nkx2-1-Cre[Tg];R26R-YFP[fl/fl]) and littermate Mtg8 mutant (Mtg8[−/−];Nkx2-1-Cre;R26R-YFP[fl/fl]), Lhx6 control (Lhx6[+/+];Nkx2-1-Cre[Tg];R26R-YFP[fl/fl]) and littermate Lhx6 mutant embryos (Lhx6[−/−];Nkx2-1-Cre;R26R-YFP[fl/fl]). Each embryo sample was processed separately for transcriptomic analysis. Dissection was performed in ice-cold Hanks' Balanced Salt Solution (HBSS, Thermo Fisher Scientific (Life Technologies, #14025050). The tissue was digested with Accutase (Sigma Aldrich, #A6964) supplemented with 10 μg /ml DNAse I (Sigma Aldrich, #D5025-15KU) and dissociated by gentle trituration, before centrifugation and resuspension Neurobasal medium (Thermo Fisher Scientific (Life Technologies), #21103049). To obtain a single-cell suspension, dissociated tissue was filtered using cell strainers (Fisher Scientific Ltd, Bel Art Products, #15342931). YFP[+ve] cells were sorted using MoFlo XDP Cell Sorter. Cells were excited with a 488 nm laser, and fluorescence was detected with 530/40 nm and 580/30 nm bandpass filters. Single, whole cells were isolated from debris based on side scatter and forward scatter parameters. Background fluorescence was determined based on the non-fluorescent control samples, and cells from the upper 80% of non-background events were collected as positive fluorescent cells. Fluorescent cells were sorted directly into tubes prefilled with 300 μl Neurobasal medium. Total RNA was isolated from freshly sorted cells without freezing using RNeasy Plus Micro Kit (Qiagen, #74034) according to manufacturer's manual. RNA quality was assessed using TapeStation RNA ScreenTape (Agilent) according to manufacturer's manual.

### RNA sequencing and analysis

The NEBNext Single Cell/Low Input RNA Library Prep Kit was used to generate cDNAs from total RNA isolated from fluorescent cells that had been sorted from control and mutant embryonic cortices. The cDNA was used to prepare libraries using the Nextera XT kit (Illumina). These were run on a NextSeq 500 sequencer (Illumina). Raw sequence data were adapter and quality trimmed (fastp v0.20.1), aligned to the genome with STAR (v2.7b) and reads per transcript determined with FeatureCounts (v1.6.4). Sex-linked genes were filtered out (using GTF version M23 GRCm38.p6 from GENECODE) to eliminate the effect of uneven sex distribution within control and mutant samples which was skewing the data. Differentially expressed genes were identified using DeSeq2 in R and the two-tailed Wald test which calculates p-adj using the BH (Benjamini-Hochberg) correction for multiple comparison. For volcano plots the $log_2$ fold change cut-off was 0.5. Heatmaps and volcano plots were created using GraphPad Prism software.

### Cell Transfection and Protein Extract Preparation

Cos7 cells were cultured in DMEM (Thermo Fisher Scientific, #31966021) supplemented with 10% FBS (Sigma, #F9665), L-Glutamine

(Thermo Fisher Scientific, #25030024), MEM Non-Essential Amino Acids Solution (Thermo Fisher Scientific, #11140050) and Penicillin-Streptomycin (Thermo Fisher Scientific, #15140148). 24 h before transfection, cells were plated at a density of 300,000 cells/well in 6-well plates and transfected with 1.5–1.0 μg of each plasmid DNA using Fugene HD Transfection Reagent (Promega UK Ltd, #E2311) at a plasmid DNA:Fugene HD Transfection Reagent ratio of 1:4, according to manufacturer's manual. 72 h post-transfection, cells were collected in the presence of protease inhibitors (Roche, #589279001) and DNAse I (25 μg /ml, Sigma Aldrich, D5025-15KU) and sonicated in an ice bath 3×7"ON;30"OFF, amplitude 6, using SoniPrep 150 Plus sonicator. Next, cells were incubated on ice for 25 minutes and centrifuged at 4 °C for 20 min. Supernatants (protein lysates) were collected in pre-cooled 1.5 ml low protein binding collection tubes (Thermo Fisher Scientific-Life Technologies, #90410) and pellets were discarded. Plasmids used for co-IP experiments are indicated in Supplementary Data 3.

### Co-immunoprecipitation (Co-IP)

Per reaction, 30 μl magnetic beads (Thermo Fisher Scientific, #88847-for HA Tag antibody and mouse IgG control or #88846-for FLAG antibody and rabbit IgG control) were blocked for 30 min in 500 μl 2% BSA in PBS at 4 °C. After 30 min, blocking solution was discarded and a fresh aliquot of 500 μl PBS with 2% BSA was added to the beads. Depending on the assay, 2 μg of HA antibody (Thermo Fisher Scientific, #26183), 2 μg of Mouse IgG Isotype Control (Thermo Fisher Scientific, #14-4714-82), 0.5 μg of FLAG antibody (Cell Signaling, #2368) or 0.5 μg Rabbit IgG Isotype Control (Thermo Fisher Scientific, #31235) was used per reaction. Beads were incubated with antibodies/or/IgG controls O/N at 4 °C. The next day, beads were washed twice with 500 μl ice-cold lysis buffer (20 mM Hepes pH 7.4, 100 mM KCl, 2 mM MgCl2, 1% Triton X-100) supplemented with protease inhibitors (Roche, #589279001). 720 μl fresh protein lysate was added to the beads and incubated Oovernight at 4 °C on a rotating wheel and the remaining 80 μl were used as Input samples. The next day, beads/antibody/protein complexes were washed 3x with 500 μl ice-cold lysis buffer (20 mM Hepes pH 7.4, 100 mM KCl, 2 mM MgCl$_2$, 1% Triton X-100). Protein complexes were eluted from the beads with 80 μl 1x Loading dye (New England BioLabs, B7703S) solution at 55 °C for 10 min.

### Western Blotting

Loading dye and DTT (New England BioLabs, #B7703S) were added to the samples to a final concentration of 1x prior to denaturation. Samples were denatured for 10 minutes at 95 °C. Protein transfer was done using a semi-dry protein transfer apparatus for 1 h at 16 V using CAPS/MeOH transfer buffer (20% MeOH, 0.01 M CAPS, 0.3 M Tris, 0.08% SDS). Membranes were blocked for 1 hour at room temperature in 5% non-fat milk-TBST (20 mM Tris, 150 mM NaCl, 0.1% Tween-20, pH 7.6). Primary antibodies were diluted in in 5% non-fat milk-TBST (20 mM Tris, 150 mM NaCl, 0.1% Tween-20, pH 7.6). Membranes were incubated with primary antibodies Oovernight at 4 °C. The following primary antibodies were used: HA (1:1000, Thermo Fisher Scientific, #26183), FLAG (1:1000, Cell Signaling, #2368), MYC (1:1000, #M4439, Sigma-Aldrich), LHX6 (1:1000, custom-made), MTG8 (1/2000, #SC9737, Santa Cruz). The next day, membranes were washed 3 × 15 min in TBST. Secondary antibodies were diluted 1:2500 in in 5% non-fat milk-TBST (20 mM Tris, 150 mM NaCl, 0.1% Tween-20, pH 7.6) and incubated with the membranes for 1 h at room temperature. Next, membranes were washed 3×15 minutes in TBST. The following secondary antibodies were used: IPKine HRP, Goat anti-Mouse IgG LCS (Insight Biotechnology, Abbkine Scientific, #A25012-1), Anti-Goat IgG EasyBlot anti-Goat IgG, HRP (Insight Biotechnology, GeneTex International Corporation, #GTX628547-01), IPKine HRP, Mouse anti-Rabbit IgG LCS (Insight Biotechnology, Abbkine Scientific, #A25022-1). Blots were developed using Immobilon Forte Western HRP substrate. Uncropped and unprocessed scans are shown in the Source Data file.

### Ex-utero head electroporation

Ex-utero head electroporation was carried out on E14.5-E15.5 embryos. Briefly, embryos were dissected in ice cold KREBS buffer and ~5-10 μg of DNA diluted in TE buffer containing fast green dye was injected into the ventricle (Femptojet, Eppendorf). DNA was electroporated into the cortex by applying 5 pulses of 50 V current, with 50 ms duration and 1 s interval (BTX ECM 830 generator) using Platinum Tweezertrode 3 mm Diameter electrodes. Brains were dissected into KREBS buffer and sectioned into 250 μm slices. Slices were cultured on Nuclepore 8 μm Whatman membranes (SLS, #110414) floating in Neurobasal medium in Organ tissue culture dishes (BD Biosciences Falcon, #353037) for 48 hours before being processed for ISH or IHC. Plasmids used for electroporation are indicated in Supplementary Data 3.

### Quantification, Statistics and Reproducibility

Experiments presented in this study were repeated independently a minimum of two times. For all quantification experiments a minimum of three mice were used for each genotype. Precise n numbers used in each experiment is specified in figure legends. All embryonic sections used for quantification had a thickness of 20 $\mu$m and counts were performed within that thickness. Sections from adult mice had a 30 μm thickness, with the exception of sections from transplantations, which had a thickness of 50 μm. For each adult mouse or embryo used for counting, numbers were generated by averaging quantifications from 3–5 consecutive sections at similar anterior-posterior or Bregma levels. Cell counts were performed by investigators blind to the genotype. Statistical analysis was performed in GraphPad Prism. All data were tested for normality using a Kolmogorov–Smirnov or Shapiro–Wilk tests and subsequently analysed using an appropriate statistical test: two-tailed unpaired $t$ test with Welch's correction, one-way or two-way ANOVA with post hoc uncorrected Fisher's Least Significant Difference (LSD) test for normally distributed data; and the nonparametric two-tailed Mann–Whitney $U$ test for non-normally distributed data, unless specified otherwise. All $t$ tests were two-tailed with an alpha of 0.05.

### Reporting summary

Further information on research design is available in the Nature Research Reporting Summary linked to this article.

## Data availability

All data generated or analysed during this study are included in this published article and its supplementary information files. Source data are provided with this paper. Data from our transcriptomic analyses have been deposited to the Gene Expression Omnibus (GEO) database under accession numbers GSE195743 and GSE207506. Source data are provided with this paper.

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

## Acknowledgements
We thank our colleagues at the Wolfson Institute for Biomedical Research (University College London) for helpful comments and discussions. Matthew Grist provided excellent technical assistance. We are grateful for the IMPC and the Mary Lyons Centre for generating the floxed *Mtg8* mice and the UCL Biological services facility for outstanding animal care. We thank the UCL Genomics facility for providing

Microarray and RNAseq services and Dr Andrei Okorokov for advice on co-IP. We thank Dr Matthew Trotter for initial help with analysis of Microarray data. We also thank the B.S.R.C. "Alexander Fleming" Animal House and Bio-Imaging Facilities. M.G.O was funded by a PhD studentship from the Portuguese Fundação para a Ciência e Tecnologia (SFRH/BD/69008/2010). A.N.R was funded by a PhD studentship from the UK Wellcome Trust. Financial support to MD was provided by the Hellenic Foundation for Research and Innovation (ID 2564), Fondation Santé, and a Stavros Niarchos Foundation grant to B.S.R.C. "Alexander Fleming", as part of the Foundation's initiative to support the Greek research centre ecosystem. Financial support for the work was provided to NK by the European Research Council (Grant agreement 207807), the UK Biotechnology and Biological Sciences Research Council (BB/N009061/1) and the UK Wellcome Trust (108726/Z/15/Z).

## Author contributions

Conceptualization, Z.A., M.G.O., A.S., V.P., S.W.H., M.D., N.K.; Investigation, Z.A, M.G.O., A.S., I.M., A.N.R., L.M., M.G., M.D., N.K. Writing original draft N.K. Editing, all authors. Funding acquisition, N.K.

## Competing interests

The authors declare no competing interests.
