## [Peer Review File · Nature Communications]

Title: MTG8 interacts with LHX6 to specify cortical interneuron subtype identityREVIEWER COMMENTS

Reviewer #1 (Remarks to the Author):

Review of kessarlis MTG8 Lhx6 22

This paper examines cerebral cortical interneuron fate determination, in particular fates of interneuron progenitors in the medial ganglionic eminence, in which Lhx6 has been established to influence both migration the fate of both parvalbumin and somatostatin subclasses, although mechanisms by which it differentially influences these fates is not known. The authors convincingly demonstrate that MTG8, possibly by direct binding to LHX6 protein, directs interneuron subclass fate determination into an SST+ subclass that also expresses NPY. They also show that early expression of MTG8, probably very shortly after cell cycle exit, is required for the fate effects whereas later expression during the migratory phase is dispensable. This paper is an important step in understanding cortical interneuron subclass fate determination, a process that is critical for normal cortical development.

Comments are essentially minor:

- 1) SST+ striatal interneurons also affected by the MTG8 conditional KO with Nkx2.1?
- 2) The authors suggest that MTG8 function is via its interaction with Lhx6. However, the data does not exclude the possibility that MTG8 could even work both in parallel but also upstream of Lhx6. With the MTG8 LacZ knockin, it would thus be useful to demonstrate that MTG8 is restricted to postmitotic cells (i.e. via colabeling with KI67). The expression pattern suggests it begins to express in the SVZ which has a mix of migratory/postmitotic as well as mitotic progenitors. It could also be useful to use their system to re-evaluate whether Lhx6 protein expression initiates during the final cell cycle or after.
- 3) It is respected that detailed counts of all cortical areas are outside the scope of this paper, but some mention of for example hippocampal SST interneurons would be warranted.
- 4) How is Sox6 protein expression in migrating interneurons affected by the MTG8 KO?
- 5) what are the morphology, connectivity or intrinsic ephys characteristics of MTG8 mutant interneurons (could need SST-Cre cKO and floxed allele)? This may be beyond the scope of the current paper, but the authors should at the minimum discuss the possibility that MTG8 is required for SST/NPY expression per se but not necessarily other subclass-selective aspects of these cells (inputs, outputs, firing properties).
- 6) Easier—what about GABA and Sox6 expression by transplanted Mtg8 KO MGE cells. Does upregulation of Emx1 suggest some hybrid fates (Lhx6+ pallidal, Emx1+ pallial)?
- 7) Given the very small effect size of altering SST or PV expression in the MTG8 overexpression study, it

should be mentioned that rather than altering PV versus SST fate the overexpression could reduce viability of PV cells.

8) The paragraph in lines 170-180 mislabels reference to fig 3 panels as fig 4 panels.

Reviewer #2 (Remarks to the Author):

The manuscript by Asgarian et al, entitled “MTG8 interacts with LHX6 to specify cortical interneuron subtype identity” describes the role of the transcriptional co-factor MTG8 in the specification of a subset of MGE-derived SST-interneurons. The manuscript is an important contribution to the answer of how cortical interneuron identity and specification is established and offers further understanding to interneuron fate specification. To date, despite enormous progress being made, the specification of SST-interneurons remains unclear. While the contribution of *lhx6* to SST interneuron specification has been studied by several groups (including the authors’ groups) and in several follow-up studies (e.g. Liodis et al., 2007; Zhao et al., 2008; Neves et al., 2013; Liu et al., 2019), a new aspect of SST interneuron specification is the combined action of *lhx6* together with *mtg8* that the authors propose in the present manuscript.

Using a great variety of experimental approaches, including in situ hybridization, generation of *mtg8* conditional knockouts in MGE-derived neurons, transplantation of MGE-derived neurons from *mtg8* KOs into the cortex of WT postnatal pups, Co-IP, and gene expression analysis, the authors show that 1) the mRNA expression pattern of *mtg8* resembles that of *lhx6*, 2) loss of *mtg8* results in reduced numbers of SST mRNA in the developing cortex, 3) transplantation of MGE precursors into the cortex of WT pups shows reduced SST+ cells numbers in the cortex, 4) *sst* and *npy* are the top commonly down-regulated genes in *mtg8* and *lhx* mutants, 5) *mtg8* acts autonomously to promote SST fate and in combination with *lhx6* to promote SST+/NPY+ interneuron fate and 6) LHX6 and MTG8 proteins physically interact with each other.

I have, however, some concerns regarding wordings, methods and conclusions of this study and suggest a revision of the manuscript where necessary.

Please find my concerns listed below:

- 1) The M&M parts lacks a description of cell counting methods. From which area of the prospective cortex or from the MGE were the data derived and which volume of brain was analyzed? How were the cell numbers in Fig. 2e, Fig. 3d, extended Fig. 1 and extended Fig. 2 assessed (quantifications missing)? This needs to be clearly addressed in the M&M section.
- 2) In ll. 120-125, the authors state that they found a clear decrease in SST interneuron numbers in transplanted cells from *Mtg8*^{-/-} embryos after ‘taking cell death into account and normalizing for cell loss’. Could the authors please specify how this was achieved?
- 3) Given that the cortex has not formed by P0, the authors should rather refer to the developing or prospective cortex but not plainly to the cortex (e.g. ll. 77, 82, 83, 96 etc.).

- 4) In order to being able to compare the data derived from Fig. 3 c/d (E14.5-15.5 + 48 hours) with that of Fig. 3 e/f (P0 + 40 days), it would have been nice had the authors cultured the ex utero transplanted cortical slices for longer than 48 hours as SST cells numbers only reach plateau levels after the second postnatal week. Therefore, the comparison of prenatal sst mRNA comparison with P40 protein levels is somewhat skewed. In addition, could the authors please specify which area of the (prospective) cortex did they analyze in either type of sections? This should be addressed in the discussion.
- 5) I think that in line 176, the authors want to refer to Fig. 3e and not 4e? Likewise line 180, Fig. 3f?
- 6) In l. 153, the authors mistakenly refer to Fig. 1 instead of extended Fig. 1.
- 7) The authors should please check for consistency regarding spaces after unit characters (e.g. ll. 593, 594-595, 602 etc).
- 8) Line 490, the word Hoechst 33258 contains transposed letters (Hoescht33258).

Reviewer #3 (Remarks to the Author):

The manuscript by Asgarian et al., reveal a new molecular mechanisms in the development of cortical interneurons. The authors identify that MTG8 is a new marker and developmental regulator of the SST-NPY interneuron subtype generated from the MGE. The authors also show that MTG8 physically interacts with LHX6 one of the key regulators of early cortical interneuron development. This interaction is independent of the LHX6 LIM domains and the LIM interacting protein LDB1. The authors complete a thorough phenotyping analysis of the MTG8 null forebrain development including the analysis of ventral progenitors, patterning genes, cortical interneurons, cortical projection neurons, and striatal neurons using in situ hybridization on tissue, RNA sequencing of MGE cells, and control/mutant MGE cell transplant studies. Finally, the authors address cell type specific and timing roles for MTG8 during cortical interneuron development by generating a conditional KO for MTG8 and using Nkx2.1-cre for early deletion and Lhx6-cre for later deletion. Interestingly, only the Nkx2.1-cre driven cKO showed the SST phenotype at adult stages (P40) suggesting an early role for MTG8 in SST cortical interneuron development. The manuscript is well written and the results are important for the cortical interneuron field. The authors reveal that MTG8 is a new regulator of SST+ interneurons and provide new mechanisms of cortical interneuron development from the MGE. I identified some major and minor points outlined below for the authors to address in the manuscript.

MAJOR points

1. The authors could provide more details on how they discovered the enriched expression of Mtg8 and Mtg16. They only mention:
“comparative transcriptomic analysis of purified embryonic MGE- and CGE-derived cortical interneurons (data not shown)”
2. Knock-in reporter alleles can be more stable compared to the endogenous protein. Does β -gal expression overlap 100% with MTG8 protein expression or do β -gal+ only cells exist during MGE/cortical interneuron development at early and late embryonic stages? The authors use both β -gal (Fig 1) and

MTG8 protein expression (Ext Data Fig. 1) so this is an important point to address in characterizing MTG8 cells. If the β -gal is more stable, it could be used as a short-term fate-map.

3. Can the authors clarify if the SST reduction in MTG8 null embryos is specific to cortical areas or also in the ventral forebrain? The ext data figure 1J panel shows the more severe reduction is in the migratory cells in the cortex compared to the ventral forebrain.

4. The authors show that progenitor cells, patterning, and LGE differentiation markers are unchanged in MTG8 null embryos using impressive array of gene expression. Can the authors describe in more detail (or show additional pictures) if the dorsal and ventral axon tracts are forming in the MTG8 null embryos? The authors only mention that the anterior commissure is missing in null embryos (lines 113-114).

5. The authors state early in the results (line 153-155) about MTG8 and LHX6 “the two genes are activated independently of each other through parallel pathways” Lhx6 expression is normal in MTG8 null embryos, however, MTG8 conditional KOs generated with Nkx2.1-cre show a reduction of Lhx6 expression in the postnatal cortex. Can the authors clarify this difference? Does MTG8 play different roles depending on the timing of cortical interneuron development and Lhx6 expression? It would be helpful if the authors could expand on this in the discussion.

6. The MTG8 cKO with Nkx2.1 cre was described at P40 in Figure 4. The kinetics of the phenotype remain largely unclear. In addition, there is no direct comparison of embryonic MTG8 cKO to the germline null embryonic phenotype. Examining SST and LHX6 expression at early postnatal stages in the cKO would also be an important comparison to the P40 phenotype.

MINOR points:

7. Figure 3F – What is the percent increase in MTG8 overexpressing compared to control for SST neurons? The authors show an asterisk for significance in the graph but it seems important to specify what is the actual difference in this experiment? Also, the labeling is incorrect in the results section lines 172-181, where it refers to these results in Fig. 4 when it should be Fig. 3.

8. The authors mention that MTG16 null mice develop seizures at adult stages but the cellular/molecular phenotype is not clear. Do the MTG8 cKO mice develop similar adult phenotype?

9. The authors show SHH expression in panel O in extended data figure 2. However, this does not look like expected SHH expression. SHH is expressed in the MGE/ventral regions as shown in panel G. However, panel O shows clear expression in LGE and Septum.

10. Is the most updated gene nomenclature for mouse- Runx1t1 or Mtg8?

REVIEWER COMMENTS and OUR RESPONSES

We thank the reviewers for their enthusiasm, their thorough assessment of the manuscript and their insightful and helpful comments. We addressed all concerns either experimentally or through discussion and clarification. We hope the reviewers will find the revised manuscript significantly improved. A point-by-point response follows:

Reviewer #1 (Remarks to the Author):

1. SST+ striatal interneurons also affected by the MTG8 conditional KO with Nkx2.1?
We quantified striatal and hippocampal Sst and Pv interneurons and, similar to the phenotype in the cortex, we find reduction of Sst but not Pv cells in both regions (see Supplementary Fig. 9b-c). Within the striatum, SST/NPY but not SST/nNOS subsets are missing (Supplementary Fig. 9c), matching the cortical phenotype.
2. The authors suggest that MTG8 function is via its interaction with Lhx6. However, the data does not exclude the possibility that MTG8 could even work both in parallel but also upstream of Lhx6. With the MTG8 LacZ knockin, it would thus be useful to demonstrate that MTG8 is restricted to postmitotic cells (i.e. via colabeling with KI67). The expression pattern suggests it begins to express in the SVZ which has a mix of migratory/postmitotic as well as mitotic progenitors. It could also be useful to use their system to re-evaluate whether Lhx6 protein expression initiates during the final cell cycle or after.
We now discuss clearly the possibility that MTG8 may be acting upstream or downstream of LHX6 within the SVZ/mantle of the MGE and why we propose that the two proteins interact to mediate their function (see paragraph 3 in the discussion). Unfortunately, in our hands, neither the LHX6 nor the MTG8 antibodies are robust enough to detect low levels of protein which might be present as the cells emerge from the VZ. For this reason, we could not finely dissect their expression in the different zones of the MGE. Of note, single cell transcriptomic studies have reported enriched expression of *Mtg8* in postmitotic immature GABAergic neurons (<https://doi.org/10.1126/science.aar6821>, Figure 1 and <https://doi.org/10.1038/srep45656>, Figure 1). It remains to be confirmed whether this expression starts at the final stages of the cycle or after cell cycle exit.
3. It is respected that detailed counts of all cortical areas are outside the scope of this paper, but some mention of for example hippocampal SST interneurons would be warranted.
Please see point 1 above. We now quantify hippocampal and striatal interneurons in the mutants. The data are shown in Supplementary Fig. 9b-c.
4. How is Sox6 protein expression in migrating interneurons affected by the MTG8 KO?
We performed ISH for Sox6 (not shown) and immunohistochemistry for SOX6 and YFP (detecting migrating MGE cells) in control and *Mtg8* conditional mutant embryos at E13.5 (See Supplementary Figure 8c). We do not see a difference in SOX6 expression between the two genotypes. This is in line with the lack of significant changes in *Sox6* expression upon *Mtg8* loss in our transcriptomic analysis.
5. what are the morphology, connectivity or intrinsic ephys characteristics of MTG8 mutant interneurons (could need SST-Cre cKO and floxed allele)? This may be beyond the scope of the current paper, but the authors should at the minimum discuss the possibility that MTG8 is required for SST/NPY expression per se but not necessarily other subclass-selective aspects of these cells (inputs, outputs, firing properties).
These are fascinating questions and avenues for further work. We could not explore the phenotype further (in terms of physiological properties, morphology, connectivity etc) in the current manuscript. We discuss this important point in paragraph 5 of the discussion.

6. Easier—what about GABA and Sox6 expression by transplanted Mtg8 KO MGE cells. Does upregulation of Emx1 suggest some hybrid fates (Lhx6+ pallidal, Emx1+ pallial)?
We assessed expression of Emx1 in mutant embryos by in situ hybridization and could not see any visible upregulation (data not shown). Therefore, we did not explore further the possibility of hybrid fates in transplanted cells.
7. Given the very small effect size of altering SST or PV expression in the MTG8 overexpression study, it should be mentioned that rather than altering PV versus SST fate the overexpression could reduce viability of PV cells.
An important point indeed. We make this clear in the Results section Lines 197-200. It had not been our intention to imply that there is a fate switch between SST and PV.
8. The paragraph in lines 170-180 mislabels reference to fig 3 panels as fig 4 panels.
This has now been corrected, thank you.

Reviewer #2 (Remarks to the Author):

1. The M&M parts lacks a description of cell counting methods. From which area of the prospective cortex or from the MGE were the data derived and which volume of brain was analyzed? How were the cell numbers in Fig. 2e, Fig. 3d, extended Fig. 1 and extended Fig. 2 assessed (quantifications missing)? This needs to be clearly addressed in the M&M section.

Missing quantifications:

Fig. 2e: Quantification is now shown in the revised manuscript. There is a clear reduction of migrating Npy cells in the embryonic cortex of mutant Mtg8 and Lhx6 embryos and this supports our RNAseq data. Extensive quantification of NPY has also been performed by IHC in the adult cortex and striatum of Mtg8 mutants and is shown in figures 4h, 4k and the now Supplementary Fig. 9c.

Fig. 3d – this shows the results of ectopic expression of LHX6 and/or MTG8 in the developing cortex. We did not quantify cells expressing ectopic Npy in the cortex because the Npy signal is evident from the images in experiments where both MTG8 and LHX6 were introduced. Any variation in the extent of ectopic activation of Npy is dependent on the efficiency of electroporation rather than the ability of the two proteins to activate Npy. The results have been consistent in several electroporation experiments. In contrast, overexpression of either LHX6 or MTG8 alone never induced any expression of NPY. Therefore, we reasoned that there is no need to provide a quantitative result given the consistent outcome.

Supplementary Fig. 1 (now Supplementary Fig. 3 in the revised manuscript) – this shows low magnification images of the telencephalon at E13.5. Higher magnification and extensive quantification in the developing cortex are shown in Figure 1i and 1k.

Supplementary figure 2 (now Supplementary figure 4) – this figure shows a panel of 31 genes expressed in control and mutant embryos. We restricted this analysis to a qualitative assessment, looking for major changes rather than a detailed quantitative analysis.

Wherever counting was performed, we specified the area counted in the Figure legend and we now clarify how counting was performed in the materials and methods (see last section).

As for the area of the prospective cortex presented in figures of embryonic brains, we now refer to the area as the 'prospective neocortex' to distinguish it from the prospective paleocortex and hippocampus. No further attempts were made to determine which prospective neocortical area was examined because it is not easy to accurately identify specific cortical areas at these stages.

2. In ll. 120-125, the authors state that they found a clear decrease in SST interneuron numbers in transplanted cells from *Mtg8*^{-/-} embryos after 'taking cell death into account and normalizing for cell loss'. Could the authors please specify how this was achieved? When calculating SST and PV numbers as a % of transplanted cells, a 25% increase had been applied to the total number of transplanted cells in order to account for the cell death that was found in Fig 1o. SST cell numbers among transplanted cells were found to be significantly decreased both with and without a 25% correction in total cell numbers. PV cell numbers, on the other hand, appeared increased in the mutants when no correction had been applied. We reasoned that this is due to SST cell death. In order to avoid giving the false impression of re-specification of SST into PV, we had applied the correction in the original manuscript.

For clarity and simplicity, we now present the uncorrected data in the revised manuscript Fig 1p, 1q. We provide an explanation for the apparent increase in PV abundance. See second paragraph in the results section entitled 'An autonomous requirement for *Mtg8* in the specification of *Sst* cortical interneuron subsets'.

3. Given that the cortex has not formed by P0, the authors should rather refer to the developing or prospective cortex but not plainly to the cortex (e.g. ll. 77, 82, 83, 96 etc.). We now refer to it as the developing or prospective cortex throughout the manuscript wherever embryonic images are shown.
4. In order to being able to compare the data derived from Fig. 3 c/d (E14.5-15.5 + 48 hours) with that of Fig. 3 e/f (P0 + 40 days), it would have been nice had the authors cultured the ex utero transplanted cortical slices for longer than 48 hours as SST cells numbers only reach plateau levels after the second postnatal week. Therefore, the comparison of prenatal *sst* mRNA comparison with P40 protein levels is somewhat skewed. In addition, could the authors please specify which area of the (prospective) cortex did they analyze in either type of sections? This should be addressed in the discussion.

We refrain from any quantitative comparison between Fig. 3c,d and Fig. 3e,f simply because the starting progenitor cells (E13.5 cortex versus E13.5 MGE) represent quite different populations in terms of stages of differentiation. Electroporation into E13.5 prospective cortex targets only dividing VZ cells whereas lentiviral transduction of dissociated E13.5 MGE cells targets a mixture of VZ/SVZ progenitors together with a large population of immature postmitotic neurons. Hence, the susceptibility to fate-changes between the two starting populations of cells is presumably quite different.

We nevertheless repeated the electroporation experiments and cultured the slices for 7 days. The results are shown in Supplementary Fig. 6. Even after 7 days in vitro we could not see *Sst* expression upon overexpression of *MTG8* and/or *LHX6*. Therefore, other factors in the MGE must be needed for *Sst* gene activation. Unfortunately, the slices cannot really be cultured much beyond 7 days without significant merging of structures within the slice. We conclude that additional ventral telencephalic factors are needed for upregulation of *Sst* expression.

5. I think that in line 176, the authors want to refer to Fig. 3e and not 4e? Likewise line 180, Fig. 3f?
These have now been corrected, thank you.
6. In l. 153, the authors mistakenly refer to Fig. 1 instead of extended Fig. 1.
Reference to Figure 1 is correct in this case. Figure 1h shows that expression of Lhx6 is not affected in the absence of MTG8.
7. The authors should please check for consistency regarding spaces after unit characters (e.g. ll. 593, 594-595, 602 etc).
These have now been corrected, thank you.
8. Line 490, the word Hoechst 33258 contains transposed letters (Hoescht33258).
This has now been corrected, thank you.

Reviewer #3 (Remarks to the Author):

1. The authors could provide more details on how they discovered the enriched expression of *Mtg8* and *Mtg16*. They only mention:
“comparative transcriptomic analysis of purified embryonic MGE- and CGE-derived cortical interneurons (data not shown)”.
In the revised manuscript we present the data from our transcriptomic analysis which led to the identification of the *Mtg* genes (see Supplementary Figure 1 and Supplementary Table 1). Experimental details are found in the Methods section. The data have been submitted to the GEO database under accession number GSE207506.
2. Knock-in reporter alleles can be more stable compared to the endogenous protein. Does β -gal expression overlap 100% with MTG8 protein expression or do β -gal+ only cells exist during MGE/cortical interneuron development at early and late embryonic stages? The authors use both β -gal (Fig 1) and MTG8 protein expression (Ext Data Fig. 1) so this is an important point to address in characterizing MTG8 cells. If the β -gal is more stable, it could be used as a short-term fate-map.
We quantified both β -gal (Fig. 1g) and MTG8 protein (Suppl Fig. 2h) at E18.5 and found that in both cases around 12% of MGE-derived cells are labelled in the cortex. This indicates that β -gal is an accurate reporter of MTG8 expression at this stage. We did not attempt to characterize the detailed expression of MTG8 within the lineage at these stages because (a) the phenotype that we observe (namely, *Sst* cell reduction) is detected earlier than that, at E13.5, as soon as cells emerge from the mantle of the MGE (Fig. 1h-k and Supplementary Fig. 3a-b, Supplementary Fig. 8a-b), and (b) late deletion of *Mtg8* in migrating cortical interneurons using *Lhx6-Cre* has no detectable phenotype (Supplementary Fig. 9a). Therefore, we focused our attention to the early MGE where MTG8 seems to function.
3. Can the authors clarify if the SST reduction in MTG8 null embryos is specific to cortical areas or also in the ventral forebrain? The ext data figure 1J panel shows the more severe reduction is in the migratory cells in the cortex compared to the ventral forebrain.
There is a clear reduction of *Sst* cells in the ventral forebrain and this is shown in the new Supplementary Fig. 8. We now present detailed quantification of striatal *Pv*, *Sst*, SST/NPY and SST/nNOS cells at postnatal stages (see Supplementary Fig. 9c). There is a comparable reduction in the number of *Sst* cells in all three areas examined (neocortex, striatum and the hippocampus).

4. The authors show that progenitor cells, patterning, and LGE differentiation markers are unchanged in MTG8 null embryos using impressive array of gene expression. Can the authors describe in more detail (or show additional pictures) if the dorsal and ventral axon tracts are forming in the MTG8 null embryos? The authors only mention that the anterior commissure is missing in null embryos (lines 113-114).

We have looked at the corpus callosum, hippocampal and anterior commissures. Only the latter is missing. This is now mentioned in the results lines 120-122. We are exploring the anterior commissure phenotype further, but it is beyond the scope of the current manuscript.

5. The authors state early in the results (line 153-155) about MTG8 and LHX6 “the two genes are activated independently of each other through parallel pathways” Lhx6 expression is normal in MTG8 null embryos, however, MTG8 conditional KOs generated with Nkx2.1-cre show a reduction of Lhx6 expression in the postnatal cortex. Can the authors clarify this difference? Does MTG8 play different roles depending on the timing of cortical interneuron development and Lhx6 expression? It would be helpful if the authors could expand on this in the discussion.

We do not think that MTG8 has a late function in cortical interneuron development because its deletion in migrating interneurons using Lhx6-cre, has no effect on Lhx6 or Sst cell numbers. This is shown in Supplementary Fig. 9a. The reduction in Lhx6+ve cells in postnatal mutant animals (reported in Fig. 4b, 4c) is likely due to cell death of mis-specified MGE interneurons. This is in agreement with our transplantation data where we find a ~25% reduction of cells transplanted from embryonic *Mtg8* KO MGE (see Fig. 1n-o). We clarify this important point in the results Lines 218-219 and in the discussion paragraph 5.

6. The MTG8 cKO with Nkx2.1 cre was described at P40 in Figure 4. The kinetics of the phenotype remain largely unclear. In addition, there is no direct comparison of embryonic MTG8 cKO to the germline null embryonic phenotype. Examining SST and LHX6 expression at early postnatal stages in the cKO would also be an important comparison to the P40 phenotype.

We now present data from the conditional *Mtg8* mouse at early embryonic stages. These are shown in Supplementary Fig 8. The images show reduced *Sst* cells already at E13.5 in the ventral telencephalon compared to littermate controls. *Lhx6* remains unchanged. This is comparable to the germline KO mouse and consistent with the postnatal phenotype observed and quantified in the cKO in Fig. 4 and Supplementary Fig. 9.

7. Figure 3F – What is the percent increase in MTG8 overexpressing compared to control for SST neurons? The authors show an asterisk for significance in the graph, but it seems important to specify what is the actual difference in this experiment?

We now mention the % changes in SST, NPY and PV in the results section lines 193-197. While the numbers are intriguing, we didn't want to read too much into them for two reasons: firstly, the MGE cells isolated represent a mixture of mitotic and postmitotic cells, hence only a fraction of the cells can presumably be re-specified and, secondly, the increase/reduction in cell numbers may represent a mixture of re-specification and/or cell death of interneurons overexpressing MTG8. Hence, while the overexpression experiments provide support for the overall conclusion that MTG8 has a role in SST interneuron specification, the precise changes in numbers may represent a mixture of effects inherent to such transplantation experiments.

8. Also, the labeling is incorrect in the results section lines 172-181, where it refers to these results in Fig. 4 when it should be Fig. 3.

This error has now been corrected, thank you.

9. The authors mention that MTG16 null mice develop seizures at adult stages, but the cellular/molecular phenotype is not clear. Do the MTG8 cKO mice develop similar adult phenotype?

We did not assess seizure phenotypes in *Mtg8* cKO mice because we could not allow enough mutant mice to live past the first 30-40 days. Since we had no plans to study a seizure phenotype or any other behavioural phenotype, we could not expand the mouse colony to examine this. In contrast, we maintain *Mtg16* null mice as homozygous mutants and we could observe the seizure phenotype more readily.

10. The authors show SHH expression in panel O in extended data figure 2. However, this does not look like expected SHH expression. SHH is expressed in the MGE/ventral regions as shown in panel G. However, panel O shows clear expression in LGE and Septum.

This error has now been corrected, thank you. The image shows Sp8 expression and not Shh.

11. Is the most updated gene nomenclature for mouse- *Runx1t1* or *Mtg8*?

Runx1t1 and *Mtg8* are used interchangeably in the literature. All three *Mtg* gene family members (*Mtg8*, *Mtg16* and *Mtgr1*) have been given alternative names (*Runx1t1*, *Cbfa2t3* and *Cbfa2t2*, respectively). We now mention these names right at the start of the manuscript for clarity.

REVIEWERS' COMMENTS

Reviewer #1 (Remarks to the Author):

this paper has been well revised

Reviewer #2 (Remarks to the Author):

In their revision, the authors have significantly improved the manuscript and have added relevant information where necessary. The authors addressed all my concerns either by additional analyses (and descriptions thereof) or by discussing their findings in the results and/or discussion section. I have no further objections to a publication of this manuscript.

Reviewer #3 (Remarks to the Author):

The authors provide a clear point by point response to all three reviewers. They addressed in detail the major and minor points from my first review. Overall, the revised manuscript reveals a new regulator in cortical interneuron development. The data are robust. The results are significant for the field and uncover an interesting mechanism of MTG8 and LHX6 regulating the generation of SST/NPY cortical interneuron fate.

One minor point for the authors to consider adding to the results section:

The authors state an abnormality in the forming anterior commissure. It might provide additional developmental clarity if the authors could address two points in the description:

1. Can the authors clarify from Supp. Figure 4 if the globus pallidus is similar in control vs MTG8 KO?
2. In addition, is the size of the MGE (labeled by NKX2.1 expression panel A) similar in control vs MTG8 KO?

REVIEWER COMMENTS and OUR RESPONSES

We are delighted with the reviewers' comments. Where relevant, please see our responses below:

Reviewer #1 (Remarks to the Author):

this paper has been well revised

Reviewer #2 (Remarks to the Author):

In their revision, the authors have significantly improved the manuscript and have added relevant information where necessary. The authors addressed all my concerns either by additional analyses (and descriptions thereof) or by discussing their findings in the results and/or discussion section.

I have no further objections to a publication of this manuscript.

Reviewer #3 (Remarks to the Author):

The authors provide a clear point by point response to all three reviewers. They addressed in detail the major and minor points from my first review. Overall, the revised manuscript reveals a new regulator in cortical interneuron development. The data are robust. The results are significant for the field and uncover an interesting mechanism of MTG8 and LHX6 regulating the generation of SST/NPY cortical interneuron fate.

One minor point for the authors to consider adding to the results section:

The authors state an abnormality in the forming anterior commissure. It might provide additional developmental clarity if the authors could address two points in the description:

1. Can the authors clarify from Supp. Figure 4 if the globus pallidus is similar in control vs MTG8 KO?
2. In addition, is the size of the MGE (labeled by NKX2.1 expression panel A) similar in control vs MTG8 KO?

Thank you for pointing this out. Comments on both of these have been added in the Results section lines 122-123.